# Retraction-free optimization over the Stiefel manifold with application to the LoRA fine-tuning

## Abstract

Optimization over the Stiefel manifold has played a significant role in various machine learning tasks. Many existing algorithms either use the retraction operator to keep each iterate staying on the manifold, or solve an unconstrained quadratic penalized problem. The retraction operator in the former corresponds to orthonormalization of matrices and can be computationally costly for large-scale matrices. The latter approach usually equips with an unknown large penalty parameter. To address the above issues, we propose a retraction-free and penalty parameter-free algorithm, which lands on the manifold. A key component of the analysis is the convex-like property of the quadratic penalty of the Stiefel manifold, which enables us to explicitly characterize the penalty parameter. As an application, we introduce a new algorithm, Manifold-LoRA, which employs the landing technique and a carefully designed step size strategy to accelerate low-rank adaptation (LoRA) in fine-tuning large language models. Numerical experiments on the benchmark datasets demonstrate the efficiency of our proposed method.

## 1 Introduction

Optimization over the Stiefel manifold has attracted considerable attention in the context of machine learning, e.g., RNN [3], batch normalization [10], and distributionally robust optimization [8]. The mathematical formulation of this class of problems is:

$$\min_{X \in \mathbb{R}^{d \times r}} \quad f(X) \quad \text{subject to} \quad X \in \mathrm{St}(d, r) := \{X \in \mathbb{R}^{d \times r} : X^\top X = I_d\}, \tag{1}$$

where $r \leq d$ and $f : \mathbb{R}^{d \times r} \to \mathbb{R}$ is a continuously differentiable function. The most popular methods for solving (1) are retraction-based algorithms, which have been extensively studied in the context of manifold optimization [2, 23, 6]. Recently, to alleviate the possible computational burden of the retraction operator, some retraction-free methods have been developed in [19, 18, 41, 1]. The ideas in these papers are based on a combination of the manifold geometry and a penalty function for the manifold constraint, which involves an unknown but sufficiently large penalty parameter. For large-scale machine learning applications, retraction-free algorithms are preferred. However, designing retraction-free algorithms with a known penalty parameter for solving (1) remains a challenge.

Another motivation for studying retraction-free methods arises from its application in the fine-tuning of large language models (LLMs). Recently, LLMs have revolutionized the field of natural language processing (NLP), achieving unprecedented performance across various applications [33, 32]. To tailor pretrained LLMs for specific downstream tasks, the most common approach is full fine-tuning, which requires prohibitively large computational resources due to the need to adapt all model weights, hindering the deployment of large models. As a result, parameter-efficient fine-tuning (PEFT) has gained widespread attention for requiring few trainable parameters while delivering comparable

Submitted to 38th Conference on Neural Information Processing Systems (NeurIPS 2024). Do not distribute.

or even superior results to full fine-tuning. This paradigm involves inserting learnable modules or designating only a small portion of weights as trainable, keeping the main model frozen [21, 26, 44]. Among fine-tuning methods, low-rank adaptation (LoRA) [22] has become the de facto standard among parameter-efficient fine-tuning techniques. It assumes that the change in weights lies in a "low intrinsic dimension", thereby modelling the update $\Delta W \in \mathbb{R}^{d \times m}$ by two low-rank (not greater than a small integer $r$) matrices $A \in \mathbb{R}^{r \times m}$ and $B \in \mathbb{R}^{d \times r}$, i.e., $\Delta W = BA$. Since $r \ll d$, the requirements on both storage and computation are significantly reduced. Due to its decompositional nature, there is redundancy in the representation of $\Delta W$. Traditional optimization methods for LoRA do not exploit this redundancy, which consequently undermines model performance. Instead, we reformulate LoRA fine-tuning as an optimization problem over the product of Stiefel manifolds and Euclidean spaces. Therefore, we propose an algorithmic framework called Manifold-LoRA to accelerate the fine-tuning process and enhance model performance. Moreover, by exploiting projected gradients and incorporating a parameter-free penalty, the overhead that our method incurs is relatively negligible. Our contributions are as follows:

- We first prove the existence of explicit choice for the penalty parameter by establishing a strong convexity-like condition of the nonconvex penalty problem associated with the Stiefel manifold constraint. Furthermore, for the given penalty parameter, under mild conditions, we prove that the iterates of our proposed retraction-free gradient descent method eventually land on the Stiefel manifold and achieve the optimality of (1).

- Building upon the established landing theory of retraction-free and penalty parameter-free method and the AdamW framework, we proposed a new method, Manifold-LoRA, which employs a carefully designed step size strategy to accelerate the training process of fine-tuning. Compared with the conventional AdamW method, we use the penalized gradient instead of the usual gradient, and the computational overhead is negligible.

- Numerical experiments are conducted on a wide range of NLP tasks, demonstrating the efficiency of our algorithm. Specifically, compared to the vanilla LoRA, our Manifold-LoRA with half the trainable parameters not only delivers fast convergence but also yields improved generalization. In particular, Our method converges twice as fast as baseline methods on several typical datasets, including the SQuAD 2.0 dataset and the CoLA dataset.

## 1.1 Related Work

**Optimization over the Stiefel manifold.** Optimization over the Stiefel manifold has attracted lots of attention due to its broad applications. Through the use of retraction, known as the generalization of the exponential map, the Riemannian gradient descent is proposed [2, 6, 23], where all iterates lie on the manifold. When such retraction is computationally costly, the authors [19] develop a retraction-free algorithm based on the augmented Lagrangian method. More recently, by defining the constraint dissolving operator and adding a sufficiently large penalty term, the authors [41] convert the manifold constrained problem (1) into an unconstrained problem and then apply unconstrained optimization algorithms. In [1], motivated by the convergence of the Oja's flow, a landing flow, consisting of the projected gradient and the gradient of the penalty function, is developed to retraction-free method for the squared Stiefel manifold, i.e., $d = r$. All of these methods rely on an unknown penalty parameter to ensure the convergence. This motivates us to design penalty parameter-free algorithms, which could significantly reduce the need for tuning parameters in practical implementations.

**LoRA.** There are numerous variants of LoRA aiming to improve performance or reduce memory usage. AdaLoRA [46], a well-known successor, introduces the idea of adaptively adjusting the rank of different layers by incorporating an additional vector $\boldsymbol{g}$ to serve as the diagonal of a singular value matrix. This approach leverages a revised sensitivity-based importance measure to decide whether to disable entries in vector $\boldsymbol{g}$ and in matrices $A$ and $B$. A similar work, SoRA [15], adopts the same model architecture as AdaLoRA, but proposes a different way to update vector $\boldsymbol{g}$ after training. This update rule is the proximal gradient of $\mathcal{L}_1$ loss, acting as a post-pruning method. Additionally, a recently emerged method called VeRA [25] significantly reduces memory overhead while maintaining competitive performance. Based on the idea that networks with random initialization contain subnetworks that are near-optimal or optimal [17], VeRA only uses two frozen low-rank matrices shared by all layers, training scaling vectors unique to each layer. Although LoRA has gained significant popularity and various variants have been developed, the potential for efficient training through leveraging the manifold geometry to reduce redundancy has not been well-explored.

## 1.2 Notation

For a matrix $X \in \mathbb{R}^{d \times r}$, we use $\|X\|$ to denote its Frobenius norm. For a squared matrix $A \in \mathbb{R}^{d \times d}$, we define $\mathrm{sym}(A) = \frac{A + A^\top}{2}$ and use $\mathrm{diag}(A) \in \mathbb{R}^d$ to denote its diagonal part. For two matrices $X, Y \in \mathbb{R}^{d \times r}$, we use $\langle X, Y \rangle := \sum_{i=1}^d \sum_{j=1}^r X_{ij} Y_{ij}$ to denote their Euclidean inner product. For a differential function $f : \mathbb{R}^{d \times r} \to d$, we use $\nabla f(X)$ to denote its Euclidean gradient at $X$.

# 2 Retraction-free and penalty parameter-free optimization over the Stiefel manifold

In this section, we focus on the design of retraction-free and penalty parameter-free algorithms for solving problem (1). We will first present the retraction-free algorithm and then show how the penalty parameter can be explicitly determined by characterizing the landscape of the penalty function.

## 2.1 Retraction-free algorithms

Inspired by the retraction-free algorithms [19, 41, 1], we consider the following retraction-free gradient descent method for problem (1):

$$X_{k+1} = X_k - \alpha \mathrm{grad} f(X_k) - \mu X_k (X_k^\top X_k - I_d), \tag{2}$$

where $\alpha, \mu > 0$ are step sizes and the projected gradient $\mathrm{grad} f(X_k) := \nabla f(X_k) - X_k \mathrm{sym}(X_k^\top \nabla f(X_k))$. Note that the tangent space of $\mathrm{St}(d, r)$ is $T_{X_k} \mathrm{St}(d, r) := \{\xi \in \mathbb{R}^{d \times r} : X_k^\top \xi + \xi^\top X_k = 0\}$. Then, for $X_k \in \mathrm{St}(d, r)$, $\mathrm{grad} f(X_k)$ is the projection of the Euclidean gradient $\nabla f(X_k)$ to the tangent space, i.e., $\mathrm{grad} f(X_k) = \mathcal{P}_{T_{X_k} \mathrm{St}(d,r)}(\nabla f(X_k))$. Note that the term $X_k(X_k^\top X_k - I_d)$ is exactly the gradient of the following quadratic penalty function

$$\varphi(X) := \frac{1}{4} \|X^\top X - I\|^2.$$

As will be shown in our theorem, the use of the projected gradient is essential for landing on the manifold. This differs with the usual penalty method, which optimizes $f(X) + \mu \varphi(X)$ using the update $X_{k+1} = X_k - \alpha \nabla f(X_k) - \mu X_k (X_k^\top X_k - I_d)$, needs $\mu \to \infty$ to guarantee the feasibility.

## 2.2 Explicit choice for the penalty parameter

It is known that a large penalty parameter yields better feasibility [29, Chapter 17]. To make the iterative scheme (2) be penalty parameter-free, we need a careful investigation on the landscape of the following optimization problem:

$$\min_{X \in \mathbb{R}^{d \times r}} \quad \varphi(X). \tag{3}$$

It can be easily verified that problem (3) is nonconvex and its the optimal solution set is $\mathrm{St}(d, r)$. The key of obtaining an explicit formula of $\mu$ is to establish certain strong convexity-type inequality and show the gradient descent method with step size $\mu$ has linear convergence.

For any $X \in \mathrm{St}(d, r)$, let us denote $\bar{X} := \mathcal{P}_{\mathrm{St}(d,r)}(X)$. Let $X = USV^\top$ be the singular value decomposition with orthogonal matrices $U \in \mathbb{R}^{d \times r}, V \in \mathbb{R}^{d \times d}$ and diagonal matrix $S \in \mathbb{R}^{d \times d}$, then $\bar{X} = UV^\top$. Building on these notations, we demonstrate that problem (3) satisfies the restrict secant inequality (RSI) [45], which serves as an alternative to the strong convexity in the linear convergence analysis of gradient-type methods.

**Lemma 1.** *For any $X \in \mathbb{R}^{d \times r}$ with $\|X - \bar{X}\| \leq \frac{1}{8}$, we have*

$$\langle \nabla \varphi(X), X - \bar{X} \rangle \geq \|X - \bar{X}\|^2. \tag{4}$$

With the above RSI, we have the linear convergence of the gradient descent update for (3), i.e.,

$$X_{k+1} = X_k - \mu \nabla \varphi(X_k). \tag{5}$$

**Lemma 2.** *Let the sequence $\{X_k\}$ be generated by (5) with $\mu = \frac{1}{3}$. Suppose that $\|X_0 - \bar{X}_0\| \leq \frac{1}{8}$. We have*

$$\|X_{k+1} - \bar{X}_{k+1}\|^2 \leq \frac{2}{3} \|X_k - \bar{X}_k\|^2. \tag{6}$$

The proofs of Lemmas 1 and 2 can be found in Appendix B.

## 2.3 Landing on the Stiefel manifold

Building on the established linear convergence of gradient descent for problem (3), we are now able to show that the iterates generated by (2) will land on the Stiefel manifold eventually, and the limiting point is a stationary point of (1), i.e., $\mathrm{grad} f(X_\infty) = 0$.

Let us start with the Lipschitz continuity of $\mathrm{grad} f(X)$. For any $X \in \bar{U}_{\mathrm{St}(d,r)}(\frac{1}{8})$, we define $\mathcal{P}_{T_X \mathrm{St}(d,r)}(U) = U - X\mathrm{sym}(X^\top U)$ for $U \in \mathbb{R}^{d \times r}$. We first have the following quadratic upper bound on $f$ from its twice differentiability and the compactness of $\mathrm{St}(d,r)$.

**Lemma 3.** *There exists a constant $L > 0$ such that for any $X, Y \in \mathrm{St}(d,r)$, the following quadratic upper bound holds:*

$$f(Y) \leq f(X) + \langle \mathrm{grad} f(X), Y - X \rangle + \frac{L}{2}\|Y - X\|^2. \tag{7}$$

*In addition, there exists a constant $\hat{L} > 0$ such that for any $X \in \mathrm{St}(d,r), Y \in U_{\mathcal{M}}(\frac{1}{8})$,*

$$\|\mathrm{grad} f(X) - \mathrm{grad} f(Y)\| \leq \hat{L}\|X - Y\|. \tag{8}$$

By the linear convergence result in Lemma 2, we have the following bound on the feasibility error.

**Lemma 4.** *Let $\{X_k\}$ be the sequence generated by (2) with $\mu = \frac{1}{3}$ and $\|X_0 - \bar{X}_0\| \leq \frac{1}{8}$. We have*

$$\|X_{k+1} - \bar{X}_{k+1}\| \leq \frac{2}{3}\|X_k - \bar{X}_k\| + \alpha\|\mathrm{grad} f(X_k)\|. \tag{9}$$

The following one-step descent lemma on $f$ is crucial in establishing the convergence.

**Lemma 5.** *Let $\{X_k\}$ be the sequence generated by (2) with $\mu = \frac{1}{3}$ and $\|X_0 - \bar{X}_0\| \leq \frac{1}{8}$. We have*

$$\begin{aligned} f(\bar{X}_{k+1}) - f(\bar{X}_k) \leq {}& - (\alpha - (4\hat{L}^2 + 4L + 1)\alpha^2)\|\mathrm{grad} f(X_k)\|^2 + \frac{1}{2}\|X_{k+1} - \bar{X}_{k+1}\|^2 \\ & + \frac{1}{2}\left(4\hat{D}_f + 16\hat{L}^2 + 16L + 3\right)\|X_k - \bar{X}_k\|^2. \end{aligned} \tag{10}$$

From the above lemma, the one-step descrease on $f$ is related to both the gradient norm of $f$ and the feasibility error. In terms of convergence, we need both $\mathrm{grad} f(X_k)$ and $\|X_k^\top X_k - I\|$ converge to 0. The following theorem demonstrates that the retraction-free and penalty parameter-free update (2) converges.

**Theorem 1.** *Let $\{X_k\}$ be the sequence generated by (2) with $\mu = \frac{1}{3}$ and $\|X_0 - \bar{X}_0\| \leq \frac{1}{8}$. If the step size $\alpha < \frac{1}{2c_1}$ for some $c_1$ large enough, then we have*

$$\min_{k=0,\ldots,K} \|\mathrm{grad} f(X_k)\|^2 \leq \frac{1}{K}, \quad \min_{k=0,\ldots,K} \|X_k^\top X_k - I\|^2 \leq \frac{1}{K}. \tag{11}$$

The proofs of the above lemmas and theorem are presented in Appendix B.

## 3 Accelerate LoRA fine-tuning with landing

In this section, we will first clarify where the Stiefel manifold constraint comes from in the LoRA fine-tuning. Then, we will apply the above developed retraction-free and penalty parameter-free method to enhance LoRA fine-tuning.

### 3.1 Manifold optimization formulation of LoRA fine-tuning

In neural networks, the dense layers perform matrix multiplication, and the weight matrices in these layers usually have a full rank. However, when adapting to a specific task, pre-trained language models have been shown to have a low intrinsic dimension, allowing them to learn efficiently even with a random projection to a smaller subspace. One possible drawback in the current LoRA fine-tuning framework is that the low-rank decomposition $\Delta W$ into product $BA$ is not unique. Specifically, for any invertible matrix $C$, it holds that $BA = (BC)(C^{-1}A)$. Note that $BC$ shares same

column space with $B$. This suggests us optimizing the subspace generated by $B$ instead of $B$ itself. Numerous studies in the field of low-rank optimization, e.g., [7, 13, 12], investigate the manifold geometry of the low-rank decomposition and develop efficient algorithms. However, such geometry has not been explored in the LoRA fine-tuning.

To address such redundancy (i.e., the non-uniqueness of $BA$ representations), we regard $B$ as the basis through the manifold constraint and $A$ as the coordinate of $\Delta W$ under $B$. Hence, the optimization problem can be formulated as

$$\min_{A,B} \quad L(BA), \quad \text{subject to} \quad B \in \text{St}(d,r) \text{ or } B \in \text{Ob}(d,r), \quad (12)$$

where $\text{Ob}(d,r) := \{B \in \mathbb{R}^{d \times r} : \text{diag}(B^\top B) = \mathbf{1}\}$. Compared to the Stiefel manifold $\text{St}(d,r)$, the oblique manifold $\text{Ob}(d,r)$ necessitates that the matrix $B$ has unit norms in its columns, without imposing requirements for orthogonality between the columns. Problem (12) is an optimization problem over the product of manifolds and Euclidean spaces.

## 3.2 Manifold-LoRA

The retraction-free method is well-suited to address (12), simultaneously minimizing the loss function $L(BA)$ and constraint violation. To control the constraint violation, we use the quadratic penalties $R_s(B) := \|B^\top B - I\|^2$ and $R_o(B) := \|\text{diag}(B^\top B) - 1\|^2$ for the Stiefel manifold and oblique manifold, respectively. As shown in the landing theory in Section 2, we shall use the projected gradient of the loss part instead of the Euclidean gradient. For the Stiefel manifold and the oblique manifold, the respective projected gradients are

$$\text{grad}_B L(BA) = \nabla_B L(BA) - B\text{sym}(B^\top \nabla_B L(BA)) \quad (13)$$

and

$$\text{grad}_B L(BA) = \nabla_B L(BA) - B\text{diag}(\text{diag}(B^\top \nabla_B L(BA))), \quad (14)$$

where $\text{sym}(X) := (X + X^\top)/2$. Thus, the gradients of our retraction-free method for $A$ and $B$ are $\nabla_A L(BA)$ and $\text{grad}_B L(BA) + \mu \nabla R_s(B)($ or $\nabla R_o(B))$.

Note that $B$ and $A$ represent the basis and the coordinate of $\Delta W$, respectively. This results in different magnitudes and different Lipschitz constants of their gradient function. In fact, let $X = BA$. It follows

$$\nabla_A L(BA) = B^\top \nabla_X L(X), \quad \nabla_B L(BA) = \nabla_X L(X)A^\top.$$

Then,

$$\|\nabla_A L(BA_1) - \nabla L(BA_2)\| \le \|B\|_2 L_g \|A_1 - A_2\|,$$
$$\|\nabla_B L(B_1 A) - \nabla L(B_2 A)\| \le \|A\|_2 L_g \|B_1 - B_2\|,$$

where $L_g$ is the Lipschitz constant of $\nabla_X L(X)$ and $\|\cdot\|_2$ represent the matrix $\ell_2$ norm (i.e., the largest singular value). Note that the step size generally should be propositional to the reciprocal of Lipschitz constant for the gradient type algorithms [29, 5]. Hence, we schedule the learning rates for the two matrices based on their respective $\ell_2$ norms. Having prepared the above, we incorporate the AdamW optimizer [28] with our manifold-accelerated technique to enhance the LoRA fine-tuning, as presented in Algorithm 1.

# 4 Experiments

In this section, we delve into the experimental results and their detailed analysis. This discussion is structured around two principal areas: (1) the performance gain compared to other mainstream fine-tuning methods and accelerated convergence achieved through our manifold-constrained optimization approach; (2) the convergence of matrix $B$ onto the manifold, illustrated by the heat map of $B^\top B$.

**Baselines** We compare our approach against several baseline methods, including full fine-tuning, Adapter [21], BitFit [44] and LoRA [22]. The variants of the Adapter method are excluded from the baselines, as their performance are relatively similar.

**Implementation Details** Our code is based on Pytorch [31], Huggingface Transformers [40] and an open-source plug-and-play library for parameter-efficient fine-tuning opendelta [24]. The bottleneck dimension for the Adapter is set to 16 or 32, ensuring that the number of trainable parameters aligns

**Algorithm 1:** Manifold-LoRA

---

**Input:** Initial point $A_0, B_0, \mu \in \mathbb{R}, \beta_1 = 0.9, \beta_2 = 0.999, upper\_bound \geq lower\_bound > 0,$
    $\epsilon = 10^{-8}, \gamma > 0, \lambda \in \mathbb{R},$ and $k = 0.$

**while** *Stopping conditions not met* **do**

    **for** $C \in \{A, B\}$ **do**

        **if** $C = B$ **then**

            Set $g(C_k)$ according to (13) or (14) using the stochastic estimate of $\nabla_B L(B_k A_k)$
            `// Projected gradient for matrix B`

        **else**

            Set $g(C_k)$ to be the stochastic estimate of $\nabla_A L(B_k A_k)$

        **end**

    **end**

    $m(C_k) \leftarrow \beta_1 m(C_k) + (1 - \beta_1)g(C_k)$

    $v(C_k) \leftarrow \beta_2 v(C_k) + (1 - \beta_2)g_t^2(C_k)$

    $\hat{m}(C_k) \leftarrow \frac{m(C_k)}{1 - \beta_1^t}$

    $\hat{v}(C_k) \leftarrow \frac{v(C_k)}{1 - \beta_2^t}$

    $\eta(C_k) \leftarrow clip(\text{norm}_{C_k}, upper\_bound, lower\_bound)$
                                    `// Scheduling step size of matrix A and B`

    $C_k \leftarrow C_{k-1} - \eta_t(C_k)\left(\hat{m}_t(C_k)/\left(\sqrt{\hat{v}_t(C_k)} + \epsilon\right)\right) - \lambda C_{k-1}$

    **if** $C = B$ **then**

        $C_k \leftarrow C_k - \mu \nabla R_s(C_k)(\text{ or } \nabla R_o(C_k))$ `// Apply penalty gradient for matrix B`

    **end**

  **end**

  $k \leftarrow k + 1$

**end**

---

closely with that of the LoRA method and the new layers are inserted into the attention layer and feed-forward layer. The update of LoRA is scaled by a hyper-parameter $\alpha$. This value is typically left unmodified, as it is usually set as 16 or 32 and never tuned [22, 43]. The exponential moving average parameters $\beta_1$ and $\beta_2$ of AdamW [27] are set to their default values of 0.9 and 0.999, respectively. All the experiments are conducted on NVIDIA A800 GPUs. More details are presented in Appendix C.

## 4.1 Natural language understanding

We first evaluate our backbone model DeBERTaV3-base [20] on GLUE [37] benchmark containing nine sub datasets, including MNLI [39], SST-2 [36], CoLA [38], QQP [37], QNLI [35], RTE [4], MRPC [16], and STS-B [37].

Experimental results of the GLUE dataset are recorded in Table 1. It can be seen that our method is consistently superior to other baselines. Notably, for RTE and STS-B datasets, both sphere-constrained (i.e., oblique manifold-constrained) and Stiefel-constrained have an obvious performance gain even with only half the trainable parameters compared to the LoRA baseline, i.e., Sphere$_{r=8}$ and Stiefel$_{r=8}$ beat LoRA$_{r=16}$. In addition, with the help of manifold geometry, the fine-tuning process can be significantly accelerated compared to the vanilla AdamW optimizer, achieving a lower training loss, as shown in Figure 1. Particularly on the CoLA dataset presented in Figure 1a, our approach achieves the same training loss as the standard Adam optimizer but requires nearly half the number of epochs.

## 4.2 Question Answering

We conduct an evaluation on two question answering datasets: SQuAD v1.1 [35] and SQuADv2.0 [34]. Manifold-LoRA is used to fine-tune DeBERTaV3-base for these tasks, which are treated as sequence labeling problems predicting the probability of each token as the start or end of an answer span.

The main experimental results are presented in Table 2. For LoRA and our algorithms, new layers are inserted into $W_q, W_k, W_v, W_o, FC_1, FC_2$. Notably, both manifold-regularized LoRA variants consistently outperform all fine-tuning methods. Additionally, we plot the training loss, evaluation

Table 1: Results with DeBERTaV3-base on GLUE benchmark. We denote the best results in **bold**.

| Method | # Params | MNLI m / mm | SST-2 Acc | CoLA Mcc | QQP Acc / F1 | QNLI Acc | RTE Acc | MRPC Acc | STS-B Corr | All Ave. |
|---|---|---|---|---|---|---|---|---|---|---|
| Full FT | 184.42M | 90.45/**90.60** | 95.48 | 68.17 | **91.99/89.12** | 93.60 | 79.28 | 88.93 | 90.92 | 87.85 |
| Adapter | 0.61M | 90.13/90.16 | 94.86 | 69.37 | 91.38/88.46 | 93.54 | 81.87 | 89.12 | 91.52 | 88.06 |
| BitFit | 0.06M | 87.08/86.39 | 94.88 | 69.11 | 87.96/84.35 | 92.19 | 76.52 | 87.06 | 90.96 | 85.65 |
| LoRA$_{r=8}$ | 0.30M | 90.20/90.08 | 94.93 | 68.14 | 90.78/87.68 | 93.85 | 80.15 | 90.40 | 90.29 | 87.60 |
| LoRA$_{r=16}$ | 0.59M | 90.44/90.12 | 95.41 | 68.19 | 90.92/87.77 | 94.00 | 80.58 | 90.20 | 90.34 | 87.74 |
| Sphere$_{r=8}$ | 0.30M | 90.37/90.09 | 95.48 | 69.55 | 91.25/88.34 | 94.02 | 82.44 | 91.55 | 91.26 | 88.44 |
| Sphere$_{r=16}$ | 0.59M | **90.52**/90.19 | 95.64 | **70.14** | 91.46/88.65 | **94.29** | 82.16 | **91.67** | 91.59 | **88.63** |
| Stiefel$_{r=8}$ | 0.30M | 90.25/89.99 | 95.46 | 69.85 | 91.44/88.60 | 94.09 | **83.16** | 91.18 | 91.22 | 88.52 |
| Stiefel$_{r=16}$ | 0.59M | 90.26/90.28 | **95.76** | 68.92 | 91.71/89.00 | 94.10 | 82.16 | 91.10 | 91.51 | 88.48 |

exact match, and evaluation F1 scores against epochs in Figure 2. We conclude that the proposed Manifold-LoRA method achieves a 2x speed-up in training epochs compared to AdamW, while simultaneously improving model performance. We also illustrate the heat map of $B^\top B$ in Figure 3, which indicates that the matrix $B$ lands on the manifold eventually. This supports our assertion that landing on manifold enhances the performance of LoRA.

### 4.3 Natural Language Generation

The E2E NLG Challenge[30], as introduced by Novikova, provides a dataset for training end-to-end, data-driven natural language generation systems, widely used in data-to-text evaluations. The E2E dataset comprises approximately 42,000 training examples, 4,600 validation examples, and 4,600 test examples, all from the restaurant domain. We test our method on the E2E dataset using GPT-2 Medium and Large models, following the experimental setup outlined by LoRA [22]. For LoRA, we set the hyperparameters to match those specified in the original paper.

The results from the E2E dataset are recorded in Table 3, where we focus on comparing LoRA and Manifold-LoRA. The results clearly indicate that our proposed algorithm outperforms the established baselines. Also, as shown in Figure 4, the matrix $B$ resides on the manifold even at the early training stage, validating the feasibility of our method.

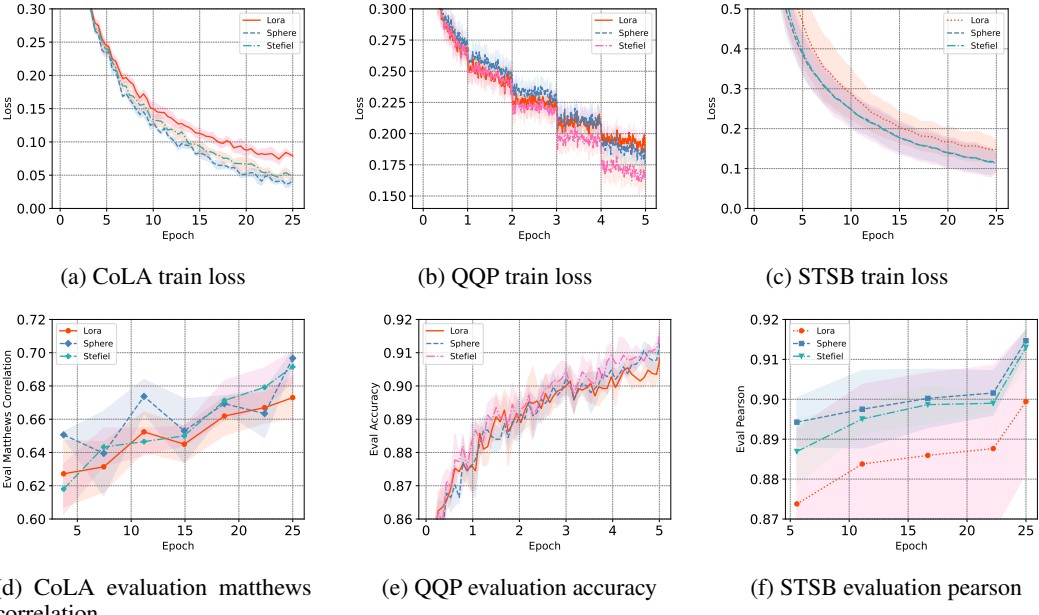

(a) CoLA train loss     (b) QQP train loss     (c) STSB train loss

(d) CoLA evaluation matthews correlation     (e) QQP evaluation accuracy     (f) STSB evaluation pearson

Figure 1: The figures illustrate that both sphere constrained and Stiefel constrained manifold-LoRA achieve a faster convergence rate and attain a lower training loss within same optimization steps compared to LoRA method on three distinct datasets CoLA, QQP, STSB.

Table 2: Results with DeBERTaV3-base on SQuAD v1.1 and SQuADv2.0. We report EM/F1. The best results in each setting are shown in **bold**.

| Methods | Params | SQuADv1.1 | SQuADv2.0 |
|---------|--------|-----------|-----------|
| Full FT | 184M | 86.30 / 92.85 | 84.30 / 87.58 |
| Adapter$_{r=16}$ | 0.61M | 87.46 / 93.41 | 85.30 / 88.23 |
| Adapter$_{r=32}$ | 1.22M | 87.53 / 93.51 | 85.42 / 88.36 |
| Bitfit | 0.07M | 80.26 / 88.79 | 74.21 / 87.19 |
| LoRA$_{r=8}$ | 1.33M | 87.90 / 93.88 | 85.56 / 88.52 |
| LoRA$_{r=16}$ | 2.65M | 87.94 / 93.75 | 85.90 / 88.81 |
| Sphere$_{r=8}$ | 1.33M | 88.51 / **94.25** | 86.33 / 89.20 |
| Sphere$_{r=16}$ | 2.65M | 88.32 / 94.03 | 86.15 / 89.03 |
| Stiefel$_{r=8}$ | 1.33M | **88.68** / 94.23 | 86.35 / 89.09 |
| Stiefel$_{r=16}$ | 2.65M | 88.25 / 94.04 | **86.41 / 89.22** |

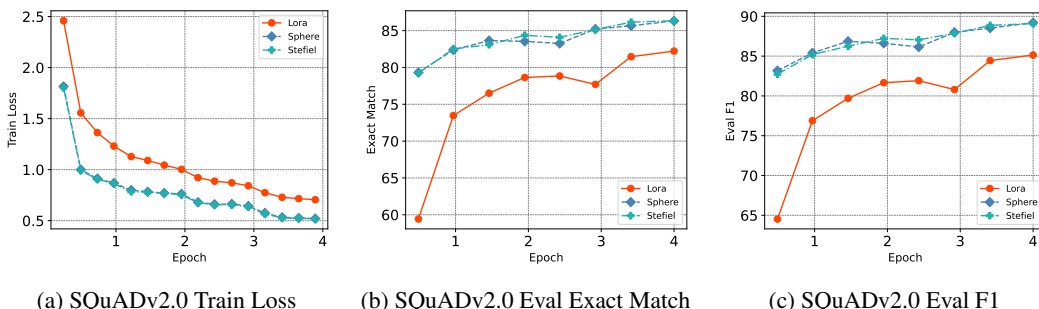

(a) SQuADv2.0 Train Loss     (b) SQuADv2.0 Eval Exact Match     (c) SQuADv2.0 Eval F1

Figure 2: The figures compare the training loss, evaluation exact match, and evaluation F1 metrics against the number of epochs for the SQuADv2.0 dataset.

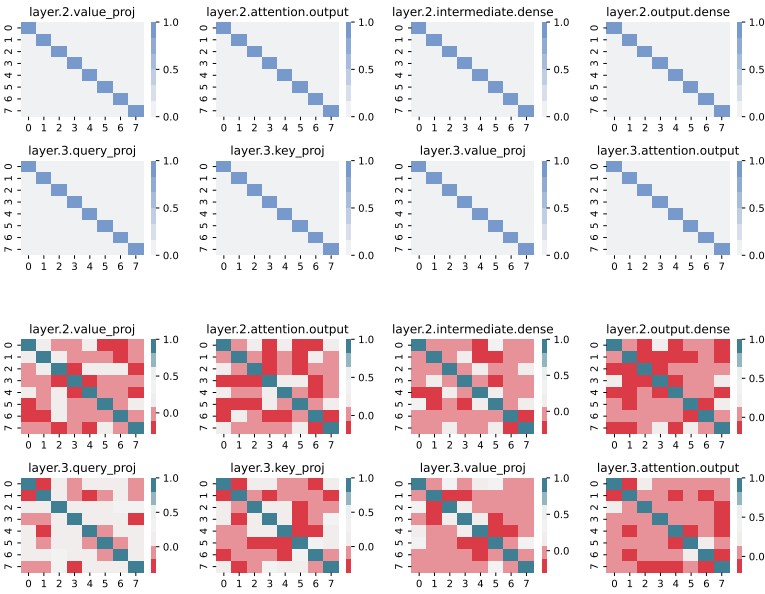

Figure 3: The heat map of $B^\top B$ with the Stiefel manifold (the first and second rows) and the oblique manifold (the third and fourth rows) at the end of training on SQuADv2.0 dataset.

Table 3: GPT-2 medium (M) and large (L) models were evaluated on the E2E NLG Challenge. * denotes results from previously published works.

| Model | Parameters | BLEU | NIST | MET | ROUGE-L | CIDEr |
|---|---|---|---|---|---|---|
| GPT-2 M (FT)* | 354.92M | 68.2 | 8.62 | 46.2 | 71.0 | 2.47 |
| GPT-2 M (Adapter$^L$)* | 0.37M | 66.3 | 8.41 | 45.0 | 69.8 | 2.40 |
| GPT-2 M (Adapter$^L$)* | 11.09M | 68.9 | 8.71 | 46.1 | 71.3 | 2.47 |
| GPT-2 M (Adapter$^H$)* | 11.09M | $67.3_{\pm.6}$ | $8.50_{\pm.07}$ | $46.0_{\pm.2}$ | $70.7_{\pm.2}$ | $2.44_{\pm.01}$ |
| GPT-2 M (FT$^{Top2}$)* | 25.19M | 68.1 | 8.59 | 46.0 | 70.8 | 2.41 |
| GPT-2 M (PreLayer)* | 0.35M | 69.7 | 8.81 | 46.1 | 71.4 | 2.49 |
| GPT-2 M (LoRA) | 0.35M | 68.9 | 8.69 | 46.5 | 71.5 | 2.51 |
| GPT-2 M(Stiefel) | 0.35M | 70.1 | 8.82 | **46.8** | **71.7** | **2.53** |
| GPT-2 M(Sphere) | 0.35M | **70.3** | **8.83** | 46.7 | **71.7** | 2.52 |
| GPT-2 L (FT)* | 774.03M | 68.5 | 8.78 | 46.0 | 69.9 | 2.45 |
| GPT-2 L (Adapter$^L$)* | 0.88M | $69.1_{\pm.1}$ | $8.68_{\pm.03}$ | $46.3_{\pm.0}$ | $71.4_{\pm.2}$ | $2.49_{\pm.0}$ |
| GPT-2 L (Adapter$^L$)* | 23.00M | $68.9_{\pm.3}$ | $8.70_{\pm.04}$ | $46.1_{\pm.1}$ | $71.3_{\pm.2}$ | $2.45_{\pm.02}$ |
| GPT-2 L (PreLayer)* | 0.77M | 70.3 | 8.85 | 46.2 | 71.7 | 2.47 |
| GPT-2 L (LoRA) | 0.77M | 70.1 | 8.82 | 46.7 | 72.0 | 2.53 |
| GPT-2 L(Stiefel) | 0.77M | 70.4 | 8.86 | **46.8** | 72.1 | 2.53 |
| GPT-2 L(Sphere) | 0.77M | **70.9** | **8.92** | **46.8** | **72.5** | **2.55** |

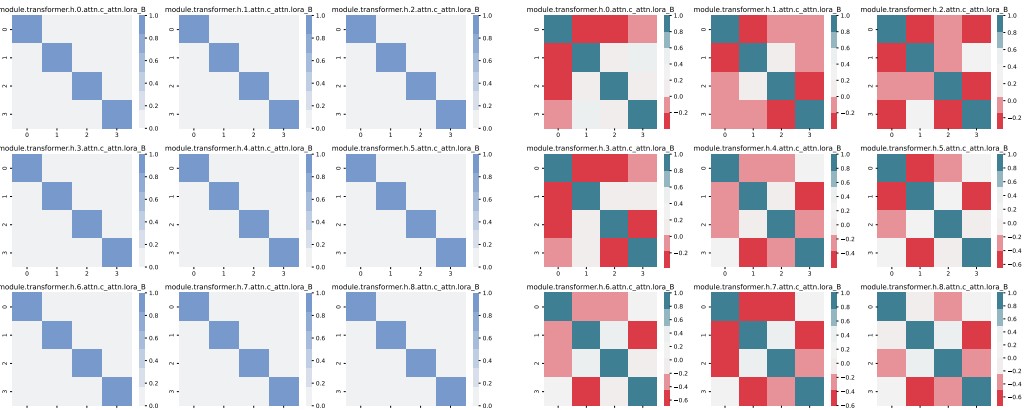

Figure 4: The heat map of $B^\top B$ with the Stiefel manifold (left) and the oblique manifold (right) on E2E dataset.

## 5 Conclusion

Optimization over the Stiefel manifold has been widely used in machine learning tasks. In this work, we develop a retraction-free and penalty parameter-free gradient method, and prove that the generated iterates eventually land on the manifold and achieve the optimality simultaneously. We then apply this landing theory to avoid the possible redundancy of LoRA fine-tuning in LLMs. Specifically, we reformulate the LoRA fine-tuning as an optimization problem over the Stiefel manifold, and propose a new algorithm, Manifold-LoRA, which incorporates a careful analysis of step sizes to enable fast training using the landing properties. Extensive experimental results demonstrate that our approach not only accelerates the training process but also yields significant performance improvements.

Our study suggests several potential directions for future research. Although the established landing theory focuses on the Stiefel manifold, extending this theory to general manifolds is one potential direction. Additionally, evaluating the performance of Manifold-LoRA on LLMs with billions of parameters would be valuable. Due to the heterogeneity of different layers, incorporating adaptive ranks for $\Delta W$ across different layers is another possible direction. This may be achievable by adding sparsity regularization to the coordinate matrix $A$.

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

## A Proximal smoothness

The notion of proximal smoothness, as introduced by [11], refers to the characteristic of a closed set whereby the nearest-point projection becomes a singleton when the point is in close enough to the set. This property facilitates algorithmic and theoretical advancements by endowing nonconvex sets with convex-like structural attributes. Specifically, for any positive real number $\gamma$, we define the $\gamma$-tube around $\mathcal{M}$ as $U_{\mathcal{M}}(\gamma) := \{x : \mathrm{dist}(x, \mathcal{M}) < \gamma\}$. We say a closed set $\mathcal{M}$ is $\gamma$-proximally smooth if the projection operator $\mathcal{P}_{\mathcal{M}}(x) := \mathrm{argmin}_{y \in \mathcal{M}} \|y - x\|^2$ is a singleton whenever $x \in U_{\mathcal{M}}(\gamma)$.

Obviously, any closed and convex set is proximally smooth for arbitrary $\gamma \in (0, \infty)$. According to [11, Corollary 4.6], a closed set $\mathcal{M}$ is convex if and only if it is proximally smooth with a radius of $\gamma$ for every $\gamma > 0$. It is worth noting that the Stiefel manifold is 1-proximally smooth. By following the proof in [11, Theorem 4.8],

$$\left\|\mathcal{P}_{\mathrm{St}(d,r)}(x) - \mathcal{P}_{\mathrm{St}(d,r)}(y)\right\| \leq 2\|x - y\|, \quad \forall x, y \in \bar{U}_{\mathrm{St}(d,r)}\left(\frac{1}{2}\right), \tag{15}$$

where $\bar{U}_{\mathrm{St}(d,r)}(\frac{1}{2}) := \{x : \mathrm{dist}(x, \mathrm{St}(d,r)) \leq \frac{1}{2}\}$ is the closure of $U_{\mathrm{St}(d,r)}(\frac{1}{2})$. It is worth noting that for any closed convex set $\mathcal{M} \subset \mathbb{R}^{d \times r}$, the projection operator $\mathcal{P}_{\mathcal{M}}$ is 1-Lipschitz continuous over $\mathbb{R}^{d \times r}$.

## B Proofs

### Proof of Lemma 1

*Proof.* Denote the SVD of $X$ by $X = USV^{\top}$. Then, it holds that $\mathrm{dist}(X, \mathrm{St}(d,r)) = \|X - \bar{X}\| = \|s - 1\|_2$, where $s = \mathrm{diag}(S)$. Furthermore, we have

$$\begin{aligned}
\langle \nabla \varphi(X), X - \bar{X} \rangle &= \langle USV^{\top}(VS^2V^{\top} - I), USV^{\top} - UV^{\top} \rangle \\
&= \langle U(S^3 - S)V^{\top}, U(S - I)V^{\top} \rangle \\
&= \mathrm{tr}((S^3 - S)(S - I)) \\
&\geq \frac{3}{2}\|s - 1\|_2^2 = \frac{3}{2}\|X - \bar{X}\|^2,
\end{aligned}$$

where the last inequality is from $\min_i s_i(s_i + 1) \geq \frac{105}{64} \geq \frac{3}{2}$. This completes the proof. $\qquad\square$

### Proof of Lemma 2

*Proof.* Assume that $\|X_k - \bar{X}_k\| \leq \frac{1}{8}$. Denote the SVD of $X_k$ by $USV^{\top}$. Let $s = \mathrm{diag}(S)$. Then, we have $\frac{7}{8} \leq s_i \leq \frac{9}{8}$ for any $i$. This implies

$$\|\nabla\varphi(X_k)\|^2 = \mathrm{tr}((S^3 - S)^2) \leq 6\|X_k - \bar{X}_k\|^2. \tag{16}$$

Hence, we have

$$\begin{aligned}
\|X_{k+1} - \bar{X}_{k+1}\|^2 &\leq \|X_{k+1} - \bar{X}_k\|^2 \\
&= \|X_k - \frac{1}{3}\nabla\varphi(X_k) - \bar{X}_k\|^2 \\
&= \|X_k - \bar{X}_k\|^2 - \frac{2}{3}\langle X_k - \bar{X}_k, \nabla\varphi(X_k) \rangle + \frac{1}{9}\|\nabla\varphi(X_k)\|^2 \\
&\leq (1 - 1 + \frac{2}{3})\|X_k - \bar{X}_k\|^2 \\
&= \frac{2}{3}\|X_k - \bar{X}_k\|^2,
\end{aligned}$$

where the first inequality is from $\bar{X}_{k+1} = \mathrm{argmin}_{X \in \mathrm{St}(d,r)} \|X - X_k\|^2$ and the second inequality is due to Lemma 1 and (16). $\qquad\square$

### Proof of Lemma 3

399 *Proof.* Due to the twice differentiability of $f$ and the compactness of $\mathrm{St}(d, r)$, the inequality (7)
400 directly follows from [9, Lemma 2.4] and [14, Lemma 4.2], where $L := L_f + D_f$ with $L_f$ being the
401 Lipschitz constant of $\nabla f(X)$ over $\mathrm{St}(d, r)$ and $D_f := \max_{X \in \mathrm{St}(d,r)} \|\nabla f(X)\|$.

402 For the second argument, we have

$$
\begin{aligned}
&\|\mathrm{grad} f(X) - \mathrm{grad} f(Y)\| \\
\leq &\|\mathcal{P}_{T_X \mathrm{St}(d,r)}(\nabla f(X)) - \mathcal{P}_{T_X \mathrm{St}(d,r)}(\nabla f(Y))\| + \|\mathcal{P}_{T_X \mathrm{St}(d,r)}(\nabla f(Y)) - \mathrm{grad} f(Y)\| \\
\leq &L_f \|X - Y\| + \frac{1}{2} \|X(X^\top \nabla f(Y) + \nabla f(Y)^\top X) - Y(Y^\top \nabla f(Y) + \nabla f(Y)^\top Y)\| \\
\leq &L_f \|X - Y\| + \frac{1}{2} \|X((X - Y)^\top \nabla f(Y) + \nabla f(Y)^\top (X - Y))\| \\
&+ \frac{1}{2} \|(X - Y)(Y^\top \nabla f(Y) + \nabla f(Y)^\top Y)\| \\
\leq &L_f \|X - Y\| + \frac{1}{2}(2\hat{D}_f + 3\hat{D}_f)\|X - Y\| \\
= &(L_f + \frac{5}{2}\hat{D}_f)\|X - Y\|,
\end{aligned}
$$

403 where $\hat{D}_f := \max_{X \in \bar{U}_{\mathrm{St}(d,r)}(\frac{1}{8})} \|\nabla f(X)\|$, the second inequality is due to the contractive property
404 of $\mathcal{P}_{T_X \mathrm{St}(d,r)}$, and the last inequality is from the fact that $\|Y\|_2 \leq \frac{3}{2}$ . By setting $\hat{L} = L_f + \frac{5}{2}\hat{D}_f$,
405 we complete the proof. $\qquad\square$

## Proof of Lemma 4

407 *Proof.* It follows that

$$
\begin{aligned}
\|X_{k+1} - \bar{X}_{k+1}\| &\leq \|X_{k+1} - \bar{X}_k\| \\
&\leq \|X_k - \mu\varphi(X_k) - \bar{X}_k\| + \alpha\|\mathrm{grad} f(X_k)\| \\
&\leq \frac{2}{3}\|X_k - \bar{X}_k\| + \alpha\|\mathrm{grad} f(X_k)\|.
\end{aligned}
$$

408 We complete the proof. $\qquad\square$

## Proof of Lemma 5

 *Proof.* It follows from (7) that

$$f(\bar{X}_{k+1}) - f(\bar{X}_k) \leq \left\langle \mathrm{grad} f(\bar{X}_k), \bar{X}_{k+1} - \bar{X}_k \right\rangle + \frac{L}{2}\|\bar{X}_{k+1} - \bar{X}_k\|^2$$

$$\leq \left\langle \mathrm{grad} f(\bar{X}_k), \bar{X}_{k+1} - X_{k+1} + X_k - \bar{X}_k \right\rangle + \left\langle \mathrm{grad} f(\bar{X}_k), X_{k+1} - X_k \right\rangle$$
$$\quad + 2L\|X_{k+1} - X_k\|^2$$

$$\leq \left\langle \mathrm{grad} f(\bar{X}_k), \bar{X}_{k+1} - X_{k+1} \right\rangle + \left\langle \mathrm{grad} f(\bar{X}_k), X_{k+1} - X_k \right\rangle$$
$$\quad + 4L(\alpha^2\|\mathrm{grad} f(X_k)\|^2 + \mu^2\|\nabla\varphi(X_k)\|^2)$$

$$= \left\langle \mathrm{grad} f(\bar{X}_k) - \mathrm{grad} f(\bar{X}_{k+1}), \bar{X}_{k+1} - X_{k+1} \right\rangle + \left\langle \mathrm{grad} f(X_k), X_{k+1} - X_k \right\rangle$$
$$\quad + \left\langle \mathrm{grad} f(\bar{X}_k) - \mathrm{grad} f(X_k), X_{k+1} - X_k \right\rangle$$
$$\quad + 4L(\alpha^2\|\mathrm{grad} f(X_k)\|^2 + \mu^2\|\nabla\varphi(X_k)\|^2)$$

$$\leq 2\hat{L}^2\|X_{k+1} - X_k\|^2 + \frac{1}{2}\|X_{k+1} - \bar{X}_{k+1}\|^2 - \alpha\|\mathrm{grad} f(X_k)\|^2 \tag{17}$$
$$\quad - \mu\left\langle \mathrm{grad} f(X_k), \nabla\varphi(X_k) \right\rangle + \frac{1}{2}(\hat{L}^2\|X_k - \bar{X}_k\|^2 + \|X_{k+1} - X_k\|^2)$$
$$\quad + 4L(\alpha^2\|\mathrm{grad} f(X_k)\|^2 + \mu^2\|\nabla\varphi(X_k)\|^2)$$

$$\leq -\alpha\|\mathrm{grad} f(X_k)\|^2 - \mu\left\langle \nabla f(X_k), \mathcal{P}_{T_{X_k}\mathrm{St}(d,r)}(\nabla\varphi(X_k)) \right\rangle + \frac{1}{2}\|X_{k+1} - \bar{X}_{k+1}\|^2$$
$$\quad + \frac{1}{2}\|X_k - \bar{X}_k\|^2 + (4\hat{L}^2 + 4L + 1)(\alpha^2\|\mathrm{grad} f(X_k)\|^2 + \mu^2\|\nabla\varphi(X_k)\|^2)$$

$$\leq -(\alpha - (4\hat{L}^2 + 4L + 1)\alpha^2)\|\mathrm{grad} f(X_k)\|^2 + \frac{1}{2}\|X_{k+1} - \bar{X}_{k+1}\|^2$$
$$\quad + (6\mu\hat{D}_f + \frac{1}{2} + 16(4\hat{L}^2 + 4L + 1)\mu^2)\|X_k - \bar{X}_k\|^2,$$

 where the second inequality is from the 2-Lipschitz continuity of $\mathcal{P}_{\mathrm{St}(d,r)}$ over $\bar{U}_{\mathrm{St}(d,r)}(\frac{1}{8})$, the third
 inequality is due to the facts that $X_k - \bar{X}_k \in N_{\bar{X}_k}\mathrm{St}(d,r)$ and $\langle A, B \rangle \leq \frac{1}{2}(\|A\|^2 + \|B\|^2)$ for any
 $A, B \in \mathbb{R}^{n \times d}$, and the last inequality comes from

$$\|\mathcal{P}_{T_{X_k}\mathrm{St}(d,r)}(\nabla\varphi(X_k))\| = \|X_k(X_k^\top X_k - I)^2\| \leq 6\|X_k - \bar{X}_k\|^2.$$

 Plugging $\mu = \frac{1}{3}$ into (17) gives (10). $\qquad\square$

 **Proof of Theorem.**

 *Proof.* Applying [42, Lemma 2] to (9) yields

$$\sum_{k=0}^{K}\|X_k - \bar{X}_k\|^2 \leq 18\alpha^2\sum_{k=0}^{K}\|\mathrm{grad} f(\bar{X}_k)\|^2 + 4. \tag{18}$$

 Then, summing (10) over $k = 0, \ldots, K$ gives

$$f(\bar{X}_{k+1}) - f(\bar{X}_0)$$
$$\leq -(\alpha - (4\hat{L}^2 + 4L + 1)\alpha^2)\sum_{k=0}^{K}\|\mathrm{grad} f(X_k)\|^2$$
$$\quad + \frac{1}{2}\left(4\hat{D}_f + 16\hat{L}^2 + 16L + 3\right)\sum_{k=0}^{K+1}\|X_k - \bar{X}_k\|^2 \tag{19}$$
$$\leq -(\alpha - (4\hat{L}^2 + 4L + 1)\alpha^2 + 9(4\hat{D}_f + 16\hat{L}^2 + 16L + 3)\alpha^2)\sum_{k=0}^{K}\|\mathrm{grad} f(X_k)\|^2$$
$$\quad + \frac{1}{2}\left(4\hat{D}_f + 16\hat{L}^2 + 16L + 3\right)(18\alpha^2\|\mathrm{grad} f(X_{k+1})\|^2 + 4).$$

418 Define $c_1 = 148\hat{L}^2 + 148L + 36\hat{D}_f + 28$ and $c_2 = (9\hat{D}_f^2 + 2)(4\hat{D}_f + 16\hat{L}^2 + 16L + 4)$. Then, we
419 have

$$\alpha(1 - c_1\alpha) \sum_{k=0}^{K} \|\mathrm{grad}f(X_k)\|^2 \leq f(\bar{X}_0) - f(\bar{X}_{k+1}) + c_2.$$

420 Therefore, for any $\alpha \leq \frac{1}{2c_1}$, taking $K \to \infty$ gives $\sum_{k=0}^{\infty} \|\mathrm{grad}f(X_k)\|^2 < \infty$. Then by (11),
421 $\sum_{k=0}^{\infty} \|X_k - \bar{X}_k\|^2 < \infty$. These lead to (11). $\qquad\square$

# C Hyperparameters

Table 4: Hyperparameter setup of Manifold-LoRA for question answering tasks.

| Method | Hyperparamter | SQuADv1.1 | SQuADv2.0 |
|---|---|---|---|
| | Warmup Ratio | 0.06 | |
| | LR Schedule | Linear | |
| | Weight Decay | 0.1 | |
| | $\beta_1$ | 0.9 | |
| | $\beta_2$ | 0.999 | |
| | Batch Size | 64 | |
| | Learning Rate | 3e-3 | |
| | Epochs | 4 | |
| Sphere(r=8) | $\mu$ | 0.85 | 0.85 |
| | Lower | 0.25 | 0.25 |
| | Upper | 0.75 | 0.5 |
| Sphere(r=16) | $\mu$ | 0.9 | 0.85 |
| | Lower | 0.25 | 0.25 |
| | Upper | 0.5 | 0.5 |
| Stiefel(r=8) | $\mu$ | 0.85 | 0.85 |
| | Lower | 0.25 | 0.25 |
| | Upper | 0.5 | 0.5 |
| Stiefel(r=16) | $\mu$ | 0.9 | 0.85 |
| | Lower | 0.25 | 0.25 |
| | Upper | 0.5 | 0.5 |

Table 5: Hyperparameter configurations of Manifold-LoRA for GLUE benchmark

| Method | Hyperparameter | MNLI | SST-2 | CoLA | QQP | QNLI | RTE | MRPC | STS-B |
|---|---|---|---|---|---|---|---|---|---|
| | Warmup Ratio | | | | 0.06 | | | | |
| | LR Schedule | | | | Linear | | | | |
| | Max Sequence Length | | | | 256 | | | | |
| | Weight Decay | | | | 0.1 | | | | |
| | $\beta_1$ | | | | 0.9 | | | | |
| | $\beta_2$ | | | | 0.999 | | | | |
| | Batch Size | | | | 32 | | | | |
| | LoRA Layer | | | | $W_q, W_v$ | | | | |
| | Epochs | 7 | 24 | 25 | 5 | 5 | 50 | 30 | 25 |
| | Learning rate | 5e-4 | 8e-4 | 5e-4 | 5e-4 | 1.2e-3 | 1.2e-3 | 1e-3 | 2.2e-3 |
| Sphere(r=16) | $\mu$ | 1 | 0.9 | 0.8 | 0.9 | 0.95 | 1.2 | 0.85 | 0.9 |
| | Lower | 0.25 | 0.25 | 0.5 | 0.5 | 0.5 | 0.5 | 1 | 1 |
| | Upper | 2 | 2 | 2 | 4 | 2 | 2 | 4 | 4 |
| Sphere(r=8) | $\mu$ | 0.95 | 0.95 | 1 | 0.9 | 1 | 0.9 | 0.85 | 1 |
| | Lower | 2 | 0.5 | 1 | 0.5 | 0.5 | 0.25 | 2 | 1 |
| | Upper | 8 | 2 | 8 | 2 | 2 | 0.5 | 4 | 8 |
| Stiefel(r=16) | $\mu$ | 0.8 | 0.85 | 0.95 | 0.9 | 0.95 | 1.2 | 0.8 | 1 |
| | Lower | 2 | 0.5 | 2 | 0.5 | 0.5 | 0.5 | 1 | 1 |
| | Upper | 8 | 1 | 8 | 4 | 1 | 2 | 4 | 16 |
| Stiefel(r=8) | $\mu$ | 0.8 | 0.95 | 0.95 | 0.9 | 0.85 | 0.9 | 1 | 1 |
| | Lower | 2 | 0.5 | 2 | 0.5 | 0.5 | 0.25 | 1 | 1 |
| | Upper | 8 | 2 | 8 | 2 | 2 | 1 | 4 | 16 |

Table 6: Hyperparameter setup of Manifold-LoRA for E2E benchmark.

| Method | Hyperparamter | GPT-2(M) | GPT-2(L) |
|---|---|---|---|
| | Warmup Steps | | 500 |
| | LR Schedule | | Linear |
| | Weight Decay | | 0.01 |
| | $\beta_1$ | | 0.9 |
| | $\beta_2$ | | 0.999 |
| | LoRA dropout | | 0 |
| | Batch Size | | 8 |
| | Learning Rate | | 2e-4 |
| | Epochs | | 5 |
| Sphere(r=4) | $\mu$ | 1 | 0.9 |
| | Lower | 0.5 | 0.5 |
| | Upper | 2 | 2 |
| Stiefel(r=4) | $\mu$ | 1 | 1.1 |
| | Lower | 0.5 | 0.5 |
| | Upper | 4 | 2 |

