# OpenReview forum: "Retraction-free optimization over the Stiefel manifold with application to the LoRA fine-tuning"
_NeurIPS.cc/2024/Conference — Submitted to NeurIPS 2024_

### Official Review · Reviewer_L45X · 2024-07-05

**Soundness:** 2
**Presentation:** 3
**Contribution:** 2
**Rating:** 5
**Confidence:** 4

**Summary:**

The authors propose a retraction-free Riemannian optimization scheme on Stiefel and oblique manifolds to perform parameter-efficient fine-tuning (PEFT) in LoRA style. The proposed approach exploits the theory of landing flows on Stiefel manifolds. Theoretical results demonstrating convergence of this iterative scheme are presented, and complemented by numerical experiments.

**Strengths:**

The proposed method combines the advantages of Riemannian methods while avoiding the computational burden of retracting on the Stiefel manifold. The application in the context of parameter-efficient fine-tuning represents a novelty, and enhances the relevance of the numerical results. The work is well-presented, covering both algorithmic aspects and the experimental section effectively.

**Weaknesses:**

All the presented theory is developed on a function defined on $St(d,r)$, while LoRA fine-tuning gives rise to an objective function $f(B,A) = L(BA)$, where $B \in St(d,r)$ (or $Ob(d,r)$), and $A \in \mathbb{R}^{r \times m}$. This objective function has to be minimized over $St(d,r) \times \mathbb{R}^{r \times m}$, and the advantages of optimizing on a compact manifold are thus lost.

Unfortunately, this makes the presented theoretical results not directly useful for the practical case under consideration.
To give a more precise statement, for example in Lemma 3, the constant $\widehat{L}$ would depend on $A$. By the mean value theorem, we would get a bound of the kind

$$
||grad_B f(A,B_1) - grad_B f(A,B_2)|| \leq C(A) ||B_1 - B_2||
$$
 (as noted in equation after line 183 for the Euclidean gradient).

 Since the space in which $A$ resides is not compact, one would need at least a uniform control on $||A_k||$ over the iterations to make the theory interesting for LoRA fine-tuning.
It is interesting to note, however, that the authors observe exact numerical convergence to the constraint in all cases.

**Questions:**

1. **Concerning the comment made in the "weaknesses" section:** In all the proofs, the Lipschitz constants $L_f$ and $D_f$ depend on $A$, making the convergence analysis not directly applicable in the numerical case under investigation. I would appreciate the authors' comment on this, maybe I am missing something here?

2. I believe however that the observed convergence behavior is not a coincidence, even if the setting is different.
The issue can be addressed by assuming $L(X) \to +\infty$ as $||X|| \to +\infty$. With enough regularity assumed, this ensures that the sublevel sets of $L$ are compact. Given an initial condition $X_0 = B_0A_0$ in the compact set $L^{-1}([-\infty,M])$, the flow $\dot A = -\nabla_A L(BA)$ decreases $L$, thereby remaining within the initial compact set $L^{-1}([-\infty,M])$. This would provide an effective bound for the Lipschitz constant, without compactness of the original domain.

3. I don't see however a way to control $||A||$ uniformly in time for "classification-like" problems, where the global interpolating minima may be off at infinity.

4. Line 93: the definition of $f$ has a typo. Also, "differentiable" is more common to use.

5. Line 131: Even if it's clear what you mean, $\bar{U}_{St(d,r)}(\frac18)$ was defined nowhere in the main manuscript.

6. The choice of notation is a bit unfortunate in some points. I would personally substitute the loss function $L$ with $\mathcal L$, just to avoid confusion with the Lipschitz constants.

I am inclined to raise my score, provided the authors clarify my doubts and address the mentioned issues (or at least discuss these points in the manuscript).

**Limitations:**

As noted in the "weaknesses" section, I believe there is a delicate point that is not addressed in the manuscript.

---

> ### Author Rebuttal · Authors · 2024-08-07
>
> Thank you for appreciating our work and providing detailed comments. The following are our responses.
>
> ### Weaknesses:
>
> All the presented theory is developed on a function defined on $\text{St}(d, r)$, while LoRA fine-tuning gives rise to an objective function $f(B, A) = L(BA)$, where $B \in \text{St}(d, r)$ (or $\text{Ob}(d, r)$), and $A \in \mathbb{R}^{r \times m}$. This objective function has to be minimized over $\text{St}(d, r) \times \mathbb{R}^{r \times m}$, and the advantages of optimizing on a compact manifold are thus lost.
>
> Unfortunately, this makes the presented theoretical results not directly useful for the practical case under consideration. To give a more precise statement, for example in Lemma 3, the constant $\hat{L}$ would depend on $A$. By the mean value theorem, we would get a bound of the kind$\lVert \text{grad}\_B f(A, B_1) - \text{grad}\_B f(A, B\_2) \rVert \leq C(A) \lVert B\_1 - B\_2 \rVert$
> (as noted in equation after line 183 for the Euclidean gradient).
>
> Since the space in which $A$ resides is not compact, one would need at least a uniform control on $\lVert A_k \rVert$ over the iterations to make the theory interesting for LoRA fine-tuning. It is interesting to note, however, that the authors observe exact numerical convergence to the constraint in all cases.
>
> **Reply:** **The variability of Lipschitz constants over an unbounded domain is not unique to our method.** Consider a simple two-layer neural network with weight matrices $U$ and $V$. For a loss function $\mathcal{L}(U, V)$, it is natural that the Lipschitz constant of $\nabla_V \mathcal{L}$ depends on the norm of $U$, where the domain of $U$ spans the entire Euclidean space, thus being unbounded.
>
> This phenomenon is common in various optimization scenarios and can be managed by introducing regularity assumptions on the loss function, such as cocoercivity and differentiability. Specifically, if the iterates of an algorithm adhere to a certain descent property on their loss functions, they will remain within a bounded level set from the cocoercivity of the loss function. We also note that there is a line of research that analyzes the convergence of algorithms by employing local Lipschitz smoothness instead of global Lipschitz smoothness.
>
> ### Questions:
>
> 1. **Concerning the comment made in the "weaknesses" section:** In all the proofs, the Lipschitz constants $L_f$ and $D_f$ depend on $A$, making the convergence analysis not directly applicable in the numerical case under investigation. I would appreciate the authors' comment on this, maybe I am missing something here?
>
> **Reply:** Gradient Lipschitz continuity is a widely used assumption in the analysis of optimization algorithms. Though this condition may seem restrictive for unbounded domains, particularly in unconstrained optimization, it can be relaxed by incorporating cocoercivity (which implies bounded level sets) and ensuring differentiability of the objective function. The key is how to ensure the iterates lie in a level set of $L$. This is easy for the deterministic setting, where the line search is an effective tool. However, this would be difficult in the stochastic setting where line search is not applicable. **It should be noted that such issue is not specific to our algorithm, but all stochastic algorithms.**
>
> 2. I believe however that the observed convergence behavior is not a coincidence, even if the setting is different. The issue can be addressed by assuming $L(X) \to +\infty$ as $\lVert X\rVert \to +\infty$. With enough regularity assumed, this ensures that the sublevel sets of $L$ are compact. Given an initial condition $X_0 = B_0 A_0$ in the compact set $L^{-1}((-\infty, M])$, the flow $\dot{A} = -\nabla_A L(BA)$ decreases $L$, thereby remaining within the initial compact set $L^{-1}((-\infty, M])$. This would provide an effective bound for the Lipschitz constant, without compactness of the original domain.
>
> **Reply:** In our response to the weakness, we acknowledge that the variability of Lipschitz constants over an unbounded domain is a common issue in machine learning tasks, not just specific to our approach. To address this, one effective strategy is to assume that the loss function is proper, coercive, and smooth. See also our responses to the weakness and Q1.
>
> 3. I don't see however a way to control $\lVert A\rVert$ uniformly in time for "classification-like" problems, where the global interpolating minima may be off at infinity.
>
> **Reply:** As in our response to previous questions, we may need to control the step size and the estimation accuracy of the gradient to ensure the boundedness of $\lVert A\rVert$ by maintaining iterations within a bounded level set. Could you please provide more details on your concerns regarding the potential blow-up of $\lVert A\rVert$?
>
> 4. Line 93: the definition of $f$ has a typo. Also, "differentiable" is more common to use.
>
> **Reply:** Revised.
>
> 5. Line 131: Even if it's clear what you mean, $\bar{U}_{\text{St}(d, r)}\left(\frac{1}{8}\right)$ was defined nowhere in the main manuscript.
>
> **Reply:** It is defined as $\bar{U}_{\rm St}(1/8) = \\{ Y \in \mathbb{R}^{n \times p} \mid \lVert Y - X \rVert_F \leq \frac{1}{8} \\}$. We have added this into the notation part.
>
> 6. The choice of notation is a bit unfortunate in some points. I would personally substitute the loss function $L$ with $\mathcal{L}$, just to avoid confusion with the Lipschitz constants.
>
> **Reply:** Thank you for the suggestion. We will revise our text accordingly to enhance readability.

---

> > ### Comment · Reviewer_L45X · 2024-08-07
> >
> > I wish, first of all, to thank the authors for their rebuttal.
> > Regarding their answers:
> >
> > **1, 2**: I agree that this issue is not specific to your proposed algorithm, it was not my intention to make it sound like a peculiarity of your work.
> > However, my impression on the manuscript is that there is a "distance" between the theory and the test cases under consideration. Of course a minimal set of assumptions is always the best, but I was expecting at least a separate result that would, in some way, also cover the cases presented in the experimental section, even if this requires making additional assumptions.
> >
> > **3**:  As mentioned before, my suggestion was to try to integrate your theoretical results more into the experimental setting. For example, in classification tasks (often of interest for your application), my concern was that the loss is of exponential type, and it is known that, for instance, on separable data, $||BA|| \rightarrow +\infty$ [1].
> > This is why I was skeptical about convergence guarantees in some cases of interest for your work (some of the GLUE Debertav3 tasks fit this setting for example).
> >
> > [1] M.S. Nacson et al., "Stochastic Gradient Descent on Separable Data: Exact Convergence with a Fixed Learning Rate", AISTATS 2019.
> >
> > In any case, I would like to increase my score, provided that you add a small paragraph discussing additional necessary assumptions for a broader class of problems (on the line of what you did in your answer).

---

> > > ### Author Response · Authors · 2024-08-08
> > >
> > > 1, 2: I agree that this issue is not specific to your proposed algorithm; it was not my intention to make it sound like a peculiarity of your work. However, my impression of the manuscript is that there is a "distance" between the theory and the test cases under consideration. Of course, a minimal set of assumptions is always the best, but I was expecting at least a separate result that would, in some way, also cover the cases presented in the experimental section, even if this requires making additional assumptions.
> > >
> > > **Reply:** The global Lipschitz continuity may not hold in certain instances. To enhance the adaptability of our convergence results for the applications detailed in the experimental section, we will add a paragraph to discuss more practical alternatives, such as coercivity, differentiability, and local gradient Lipschitz continuity [SIOPT 2023], aiming to clarify how these properties can be met in terms of the application presented in the experimental section.
> > >
> > > [SIOPT 2023] Xiaoxi Jia, Christian Kanzow, and Patrick Mehlitz. "Convergence Analysis of the Proximal Gradient Method in the Presence of the Kurdyka–Łojasiewicz Property Without Global Lipschitz Assumptions." SIAM Journal on Optimization 33, no. 4 (2023): 3038-3056.
> > >
> > > 3: As mentioned before, my suggestion was to try to integrate your theoretical results more into the experimental setting. For example, in classification tasks (often of interest for your application), my concern was that the loss is of exponential type, and it is known that, for instance, on separable data, $||BA|| \rightarrow +\infty$ [1]. This is why I was skeptical about convergence guarantees in some cases of interest for your work (some of the GLUE Debertav3 tasks fit this setting for example).
> > >
> > > [1] M.S. Nacson et al., "Stochastic Gradient Descent on Separable Data: Exact Convergence with a Fixed Learning Rate", AISTATS 2019.
> > >
> > > **Reply:** Thanks for your further clarification.
> > >
> > > In [1], the authors comment that replacing the global Lipschitz smoothness requirement with local Lipschitz smoothness and an appropriately small step size could be sufficient to keep iterates within a region where the Lipschitz constants remain uniformly bounded for the exponential-type loss function. Specifically, the footnote in [JMLR 2018, Assumption 3], a companion paper to [1], states: "The exponential loss does not possess a global $\beta$ smoothness parameter. However, initialization with $\eta < \frac{1}{\mathcal{L}(w_0)}$ ensures that the gradient descent iterates maintain bounded local smoothness."
> > >
> > > Furthermore, there is a growing body of research within the optimization community that focuses on algorithm convergence using local rather than global Lipschitz smoothness. This approach often depends on additional assumptions to ensure favorable properties of the local Lipschitz constant during iterations, such as Assumption 3.2 in [JOTA 2022], where the objective function is both bounded from below and lower-bounded by an affine function. For recent advancements, please refer to [SIOPT 2023].
> > >
> > > [JOTA 2022] C. Kanzow and P. Mehlitz. Convergence properties of monotone and nonmonotone proximal gradient methods revisited. Journal of Optimization Theory and Applications, 195(2):624–646, 2022.
> > >
> > > [JMLR 2018] Soudry, D., Hoffer, E., Nacson, M. S., Gunasekar, S. Srebro, N. . The implicit bias of gradient descent on separable data. Journal of Machine Learning Research, 19(70), 1-57, 2018.
> > >
> > > In any case, I would like to increase my score, provided that you add a small paragraph discussing additional necessary assumptions for a broader class of problems (on the line of what you did in your answer).
> > >
> > > **Reply:** Thank you for your insightful comment. We will definitely add a paragraph to discuss the potential lack of global Lipschitz smoothness and how to establish more practical assumptions for a broader class of problems, particularly in terms of the considered numerical applications.

---

> > > > ### Author Response · Authors · 2024-08-10
> > > >
> > > > Thank you for your valuable feedback. If you have any further comments, we would be happy to address them. Should there be no additional remarks, we would greatly appreciate it if you could consider raising your score.

---

### Official Review · Reviewer_yeZ7 · 2024-07-09

**Soundness:** 3
**Presentation:** 3
**Contribution:** 2
**Rating:** 5
**Confidence:** 4

**Summary:**

Retraction-free optimization algorithms on the Steifel manifold have been proposed in [1,18,19,41] etc. The motivation is that if the cost of the objective function/gradient evaluation is significantly larger than the evaluation of a retraction, then the retraction-free optimization algorithms show their advantages and efficiency. In particular, for the landing algorithm proposed in [1], the choice of the parameter in the penalty is important and may not be easy to choose. This paper gives an analysis that shows if the parameter is 1/3, the initial point is sufficiently close to the Stiefel manifold, and the step size is chosen sufficiently small, then the algorithm converges linearly to a stationary point. This result gives a concrete value of the parameter. Such a result is further merged into optimization on low-rank matrices and Manifold-LoRA is proposed. Numerical experiments show that the proposed method outperforms the baseline algorithms.

**Strengths:**

This paper gives a concrete value of $\mu$ and a gap between x_0 and \bar{x}_0 such that the algorithm converges under reasonable assumptions. Numerical experiments show that the proposed method is more effective than the existing approach.

**Weaknesses:**

(1) Though the value of mu and an upper bound of \|x_0 - \bar{x}_0\| are given concretely, the choice of step size is unknown. Theoretically, the step size needs to be sufficiently small (See Theorem 1). Any theoretical suggestion for the choice of the step size?
(2) Numerical experiments report results of the comparisons. However, the definition of ``result'' is not given. Is the result computational time or classification accuracy or a notion of correctness or something else?
(3) Problem (12) does not remove all the ambiguity. Note that if B \in St(d, r), then B A = B O O^T A = \tilde{B} \tilde{A}, where O is an orthonormal matrix and \tilde{B} = B O is still in St(d, r). Likewise for B \in Ob(d, r). Is it possible to completely remove the ambiguity by considering the quotient manifold?
(4) Why is the numerical performance of Manifold-LoRA for using Stiefel and Oblique manifold in (12) different? The optimization problem is equivalent in the sense that the local minimizer/stationary point does not change.

**Questions:**

The questions are given in the weaknesses section. More questions are given below.
(1) L117, X \in St(d, r) or X \in \mathbb{R}^{d \times r}
(2) What is the percentage of the computational cost of the retraction in the overall computations?

**Limitations:**

The limitations of the paper are discussed in the conclusion section.

---

> ### Author Rebuttal · Authors · 2024-08-07
>
> Thank you for reviewing our paper. The following are our responses.
>
> ### Weaknesses:
>
> 1. Though the value of mu and an upper bound of $|x_0 - \bar{x}_0|$ are given concretely, the choice of step size is unknown. Theoretically, the step size needs to be sufficiently small (See Theorem 1). Any theoretical suggestion for the choice of the step size?
>
> **Reply:** **It is not specific to our algorithm for unknown step size**, including retraction-based algorithms, where the step size relies on unknown Lipschitz constant of the Riemannian gradient for problem (1).
>
> **Theorem 1 demonstrates that a suitable constant step size results in convergence.** In contrast, the landing algorithm in [1] converges only to a neighborhood whose size is dependent on the step size. According to Theorem 1, an exact step size can be calculated if $L$, $\hat{L}$, and $\hat{D}_f$ are known. If these parameters are unknown, it is necessary to estimate their upper bounds or to utilize numerical strategies, such as a grid search. The role of this theory is to ensure the existence of a constant step size that allows the algorithm to converge. In our numerical experiments, we employ the grid search.
>
> 2. Numerical experiments report results of the comparisons. However, the definition of ``result`` is not given. Is the result computational time or classification accuracy or a notion of correctness or something else?
>
> **Reply:** We report the overall (matched and mismatched) accuracy for MNLI, Matthew’s correlation for CoLA, Pearson correlation for STS-B, and accuracy for other tasks. Higher is better for all metrics. Additionally, since the additional computational cost per step induced by our method is negligible, the number of epochs needed can be roughly equated to the time required.
>
> 3. Problem (12) does not remove all the ambiguity. Note that if $B \in St(d, r),$ then $B A = B O O^T A = \tilde{B}\tilde{A}$ where $O$ is an orthonormal matrix and $\tilde{B} = BO$ is still in $St(d, r)$. Likewise for $B \in Ob(d, r)$. Is it possible to completely remove the ambiguity by considering the quotient manifold?
>
> **Reply:** The distinction between the Stiefel and Grassmann manifolds is not significant, as observed in applications from low-rank optimization (e.g., matrix completion and matrix decomposition) and principal component analysis. The primary reason for choosing the Stiefel manifold over the Grassmann manifold is the ease of analytical treatment. Numerically, the differences between these two approaches are negligible, especially since the projection operators on their respective tangent spaces are identical, a common scenario in low-rank optimization and principal component analysis.
>
> 4. Why is the numerical performance of Manifold-LoRA for using the Stiefel manifold and the oblique manifold in (12) different? The optimization problem is equivalent in the sense that the local minimizer/stationary point does not change.
>
> **Reply:**
> Setting the constraint set of $B$ to either the Stiefel manifold or the oblique manifold in equation (12) leads to two distinct optimization problems, characterized by unique optimality conditions and stationary points. For the Stiefel manifold, the stationary points are defined by:
> $$
> grad f(X) - X \text{sym}(X^T\nabla f(X)) + X(X^TX - I) = 0.
> $$
> For the Oblique manifold, the condition is:
> $$
> grad f(X) - X \text{diag}(\text{diag}(X^T\nabla f(X))) + X \text{diag}(X^TX - I) = 0.
> $$
> Therefore, the sets of stationary points for the two manifolds are not equivalent. The differing geometrical constraints imposed by each manifold are expected to influence the numerical performance of the algorithm.
>
> ### Questions:
> The questions are given in the weaknesses section. More questions are given below.
>
> 1. L117, $X \in {\rm St}(d, r) ~ or ~ X \in \mathbb{R}^{d \times r}$
>
> **Reply:** It should be $X \in \mathbb{R}^{d \times r}$. Revised it.
>
> 2. What is the percentage of the computational cost of the retraction in the overall computations?
>
> **Reply:** Considering a LoRA layer represented by:
> $$
> H = (W + BA)S,
> $$
> where $H \in \mathbb{R}^{n \times k}$, $S \in \mathbb{R}^{n \times k}$, $W \in \mathbb{R}^{n \times n}$, $B \in \mathbb{R}^{n \times p}$, and $A \in \mathbb{R}^{p \times n}$, we define the gradient with respect to $H$ as $\mathcal{D}_H$. The gradients with respect to $B$ and $A$ are computed as $\mathcal{D}_H S^T A^T$ and $B^T \mathcal{D}_H S^T$ respectively. Thus, the computational cost for these gradients amounts to $\mathcal{O}(n^2k + n^2p) + \mathcal{O}(2pnk)$. Our method is retraction-free, adding only the computational tasks of projection and calculating the penalty gradient, both of which are $\mathcal{O}(np^2)$. This additional cost is negligible when $p$ is small.

---

> > ### Comment · Reviewer_yeZ7 · 2024-08-10
> >
> > Thank the authors for the replay.
> > 1) Though many algorithms do not know the constant step size, a commonly-used practical approach is using a line search algorithm such as backtracking with an appropriate initial step size to guarantee convergence. Is it possible to give a computable initial step size with a backtracking algorithm such that the result of Thm 1 holds. If it is true, then the proposed algorithm is more practical.
> > 4) I agree if only the B-subproblem is considered, then the stationary point of the subproblem is different for different constraints on B. However, Problem (12) considers L(BA) with an arbitrary A. It follows that the stationary points of L should be the same. That is what I am concerned.

---

> > > ### Author Response · Authors · 2024-08-10
> > >
> > > 1. Though many algorithms do not know the constant step size, a commonly-used practical approach is using a line search algorithm such as backtracking with an appropriate initial step size to guarantee convergence. Is it possible to give a computable initial step size with a backtracking algorithm such that the result of Thm 1 holds. If it is true, then the proposed algorithm is more practical.
> > >
> > >     **Reply:** Note that Theorem 1 differs from the standard line search-based convergence in two key aspects:
> > >     - The decrease in each step is applied not directly to the iterates, but to their projected versions.
> > >     - Each step may not always decrease the objective function value, but the accumulated effect over several steps will reduce the objective value, assuming the step size is small enough.
> > >
> > >     While this type of convergence analysis is common in optimization areas, such as, distributed optimization and min-max optimization, designing a line search method based on this approach is not straightforward. Exact convergence with a constant step size (even if the step size is unknown) is generally considered crucial, as opposed to convergence to a neighborhood or within a certain horizon.
> > >     However, it would be interesting to explore the development of a new Lyapunov function to establish a one-step decrease directly on the iterates, which could lead to a line search method suitable for this setting in future work.
> > >
> > > 2. I agree if only the B-subproblem is considered, then the stationary point of the subproblem is different for different constraints on B. However, Problem (12) considers $L(BA)$ with an arbitrary $A$. It follows that the stationary points of L should be the same. That is what I am concerned.
> > >
> > >     **Reply:** Thank you for your further clarification. By treating $BA$ as a single entity, the two problems become equivalent. Alternatively, different manifold constraints on  $B$  correspond to different parameterizations of the low-rank matrix $W$.
> > >
> > >     Although adding different constraints on $B$ (or $A$ ) may not change the stationary point to which the algorithm converges, it accelerates the convergence process by leveraging the manifold geometry, enabling more efficient movement along the manifold. This is why manifold optimization methods can be more powerful than traditional constrained optimization approaches. A similar acceleration is observed in low-rank matrix completion problems, where methods with a Grassmann manifold constraint converge faster, even though they ultimately reach the same solution with zero recovery error. Please refer to [MPC 2012, NeurIPS 2011, IEEE TIT 2012, IEEE TIT 2009] for the details.
> > >
> > >     - [MPC 2012] Wen Zaiwen, Yin Wotao, Zhang Yin; Solving A Low-Rank Factorization Model for Matrix Completion by A Nonlinear Successive Over-Relaxation Algorithm; Mathematical Programming Computation; 4 (2012), 333-361.
> > >     - [NeurIPS 2011] Boumal, Nicolas, and Pierre-antoine Absil. "RTRMC: A Riemannian trust-region method for low-rank matrix completion." Advances in neural information processing systems 24 (2011).
> > >     - [IEEE TIT 2012] Dai W, Kerman E, Milenkovic O. A geometric approach to low-rank matrix completion[J]. IEEE Transactions on Information Theory, 2012, 58(1): 237-247.
> > >     - [IEEE TIT 2009] W. Dai and O. Milenkovic, "Subspace Pursuit for Compressive Sensing Signal Reconstruction," in IEEE Transactions on Information Theory, vol. 55, no. 5, pp. 2230-2249, May 2009, doi: 10.1109/TIT.2009.2016006.

---

> > > > ### Comment · Reviewer_yeZ7 · 2024-08-11
> > > >
> > > > Thanks for the answer. I agree that manifold reformulation of an optimization problem can improve the numerical performance in the sense of convergence rate since a manifold reformulation removes ambiguity and reduces the dimension of the search space. However, the stationary points and minimizers usually remain the same as the original formulation. What I am curious about is that the effectiveness of the oblique manifold and stiefel manifold reformulations yield different effectiveness in the numerical experiments, which does not make sense to me. Is there any explanation? If yes, I think it is important to add to the paper.

---

> > > > > ### Author Response · Authors · 2024-08-11
> > > > >
> > > > > The different effectiveness in the numerical experiments in two manifold constraints owes to the following three aspects as we suggest.
> > > > >
> > > > > - The distinct manifold constraints lead to variations in the search space and in the process of ambiguity elimination.
> > > > > - Our algorithm employs an adaptive strategy to adjust the learning rate of matrix $A$, based on the observation of unbalanced Lipschitz constants. This approach aims to mitigate the asymmetry in the training process of matrices $A$ and $B$. However, the effectiveness of this adaptive estimation may vary between the two manifolds.
> > > > > - From a theoretical standpoint, the Stiefel manifold is expected to have a higher ambiguity elimination than the oblique manifold due to its richer geometric structure. Nevertheless, optimization on the Stiefel manifold is inherently more complex, which may result in lower competitive accuracy during testing in some cases.

---

> > > > > > ### Author Response · Authors · 2024-08-12
> > > > > >
> > > > > > Thank you for your valuable feedback. If you have any further comments, we would be happy to address them. Given that we addressed your primary concerns raised in the review, we would kindly ask you to adjust your review score while taking the discussion into account.

---

> > > > > > > ### Comment · Reviewer_yeZ7 · 2024-08-13
> > > > > > >
> > > > > > > I am still concerned about the ambiguity. Using the Grassmann manifold would be more intuitive since it removes all the ambiguity. Moreover, the geometry structure of the Grassmann manifold has been well-explored, and therefore optimization on such a manifold may be accelerated. At this stage, I prefer to raise the score to 5.

---

> > > > > > > > ### Author Response · Authors · 2024-08-13
> > > > > > > >
> > > > > > > > Thank you for your valuable feedback and quick response.
> > > > > > > >
> > > > > > > > We chose the Stiefel manifold over the Grassmann manifold for two main reasons:
> > > > > > > >
> > > > > > > > 1. The similar geometry between the Stiefel and Grassmann manifolds.
> > > > > > > > 2. The landing theory on the Stiefel manifold is more straightforward to establish. In contrast, the Grassmann manifold requires a more careful process to select representative elements and define the Riemannian metric (e.g., horizontally invariant metric) due to its quotient geometry.
> > > > > > > >
> > > > > > > > In the future, we plan to explore the landing theory for optimization on the Grassmann manifold and consider its potential uses in the context of LoRA.

---

### Official Review · Reviewer_ohUo · 2024-07-11

**Soundness:** 3
**Presentation:** 3
**Contribution:** 3
**Rating:** 6
**Confidence:** 3

**Summary:**

This paper considers solving optimization problems with constraints that have orthonormal columns (i.e. the matrix belongs to the Stiefel manifold). The leading method for solving such problems is the Riemannian optimization. However, Riemannian optimization requires a costly retraction operation. The authors propose to circumvent this by introducing an additional penalty terms that stirs the optimization towards respect the manifold constraints. Indeed, the authors show that with correcting setting of parameters, the optimum will be on the manifold, and the algorithm will find it. The authors advocate that an additional advantage of their algorithm is that we know how to set the parameters for the penalty term, and so their algorithm is parameter-free.

A significant part of the paper is devoted towards motivating the study in terms of low-rank adaption in LLMs, and showing experiments in that vain.

**Strengths:**

- A very elegant method for retraction free optimization on the Stiefel manifold.
- Detailed theoretical analysis showing the algorithm converges to a critical point on the manifold.
- The theoretical analysis gives explicit guidance on how to set the penalty parameter.
- The LLM applicaiton and experiments appear impressive. However, I am not an expert on this subject, so it is hard for me to asses how significant the results and evaluations are.

**Weaknesses:**

(The following were addressed by the authors in the rebuttal)

Major issues (affecting the recommendation):
1) Novelty: Citation [1] considers optimization with constraints on the orthogonal group (i.e. St(n,n)). It seems that the core idea on how to implement retraction free optimization already appears there. The authors mention this in "related work", and say that [1[ does not discuss the r<n case. Inspection of [1] reveals that this is not the entire story. In Sec 3.5 of [1] the case of r<n is discussed briefly, and it is said the results can be extended for that case. However, the authors of [1] are skeptical of the value of this, as they mention that there are fast retraction methods (i.e. Cayley) for the Stiefel manifold.
-  Setting the penalty parameter: the authors advocate that they give an explicit value for the penalty parameter. And indeed Theorem 1 sets the parameter \mu to 1/3. However, I do not think the situation is so simple. The theorems have the additional assumption that the iterates starts close to the manifold (1/8). This is, of course, easy to achieve - just start on the manifold itself. However, for the proof to work shouldn't all iterates stay inside this bound? This necessitates for the other parameter (step size) to be small enough. And indeed, the theorem requires that the step size be small enough, and does not specify how small. Without looking in detail in the proofs, my guess is that changing the penalty step size (\mu)  affects how close you need to be to the manifold (the value 1/8), which affects how small the step size need to be. In other words, the authors load all the complexity of setting the parameters onto the step size of the main objective. Saying there is  a upper bound on its value , without specifying what that value is. You cannot call this parameter free.

Another point is that the values of the parameter probably affect convergence rate, though the authors do not discuss this at all.

Minor comments (do not affect the recommendation):
- Line 117: If X is on the manifold, shouldn't \bar{X}, which is the projection of X on the manifold, be exactly X?
- Line 119: "satisfies the restricted secant"
- Line 131: What is U_St (1/8)? Not defined.  Ditto line 136.
- Line 133: If the condition of twice diff is assumed, then state it earlier.
- Eq (10): \hat{D}_f is not defined.
- Table 1: Why metrics are changing between columns?

**Questions:**

- Please address more carefully why you think you are novel with respect to [1]?
- Why are fast retraction methods not sufficient?
- How setting the parameters affect convergence rate?

**Limitations:**

Nothing to add.

---

> ### Author Rebuttal · Authors · 2024-08-07
>
> Thank you for carefully reading our manuscript and appreciating our work. The following are our responses.
> #### Novelty:
> 1. Citation [1] considers optimization with constraints on the orthogonal group (i.e., St(n,n)). It seems that the core idea on how to implement retraction free optimization already appears there. The authors mention this in "related work", and say that [1] does not discuss the $r<n$ case. Inspection of [1] reveals that this is not the entire story. In Sec 3.5 of [1] the case of $r<n$ is discussed briefly, and it is said the results can be extended for that case.
>
>     **Reply**: Our method differs with [1] from the following aspects:
>     - The construction of our landing algorithm is more straightforward. It involves the summation of the Riemannian gradient of the loss and the gradient of the penalty function of the Stiefel manifold. While the landing field in [1] shares a similar structure, its first term (as seen in Eq. (4) of [1]) is less interpretable. Our method may provide a clearer approach to design landing algorithms for general manifolds.
>     - We explore the strong convexity-like property, specifically the restricted secant inequality, of the penalty problem and provide an explicit value for the penalty parameter, which is not given in [1]. Consequently, our landing algorithm only requires tuning the step size, whereas [1] necessitates careful selection of both the step size and the penalty parameter.
>     - Our theorem demonstrates the exact convergence when using a constant step size. In contrast, the landing algorithm in [1] only converges to a neighborhood whose size depends on the step size. **This issue is acknowledged in their paper, particularly in the paragraph following Proposition 10.** Developing a theory that ensures exact convergence with a constant step size is crucial for making these methods competitive with retraction-based approaches.
>
>     Although the landing algorithm in [1] can be extended to the case where $r<n$ in Section 3.5, our analysis introduces a novel penalty parameter-free approach. Additionally, our theoretical results on exact convergence with a constant step size are both new and significantly different from those presented in [1]. We will revise our manuscript to make the comparison clearly.
>
> 2. However, the authors of [1] are skeptical of the value of this, as they mention that there are fast retraction methods (i.e., Cayley) for the Stiefel manifold.
>
>     **Reply**: Both [1] and [MP 2015] indicate that the computational cost of the Cayley transformation is $4nr^2 + \frac{40}{3}r^3$, which is more than twice the cost of our method at $2nr^2$ for any $r$. Additionally, performing a retraction on the Stiefel manifold involves a specific orthogonalization procedure that is challenging to scale and parallelize, such as the matrix inversion required in the Cayley transformation. In contrast, our landing algorithm can be executed efficiently using BLAS3 operations.
>
>     [MP 2015] Bo Jiang, and Yu-Hong Dai. A framework of constraint preserving update schemes for optimization on Stiefel manifold. Mathematical Programming 153, no. 2 (2015): 535-575.
>
> 3. Setting the penalty parameter: the authors advocate that they give an explicit value for the penalty parameter. And indeed Theorem 1 sets the parameter $\mu$ to 1/3. However, I do not think the situation is so simple. The theorems have the additional assumption that the iterates starts close to the manifold (1/8). This is, of course, easy to achieve - just start on the manifold itself. However, for the proof to work shouldn't all iterates stay inside this bound? This necessitates for the other parameter (step size) to be small enough. And indeed, the theorem requires that the step size be small enough, and does not specify how small.
>
>     **Reply**: Our results show that setting $\mu = \frac{1}{3}$and $0 < \alpha \leq \frac{1}{2c_1}$ with $c_1$ specified in Line 418 will yield convergence. To ensure that iterates remain within the designed neighborhood, the step size must not exceed $\frac{1}{24 \hat{D}_f}$, based on the derivations from Equation (9) and the norm of $\lVert\nabla f(X)\rVert$ over $\bar{U}\_{{\rm St}}(1/8)$. That is, if $\alpha \le \frac{1}{24 \hat{D}\_f}$, all iterates $X_k$ will stay within $\bar{U}\_{{\rm St}}(1/8)$ for all $k$. This step size bound is implied by the condition $\alpha \le \frac{1}{2c_1}$. We will revise our manuscript to clarify it.
>
> **For responses to the remaining questions, please refer to the general response.**

---

> > ### Comment · Reviewer_ohUo · 2024-08-13
> >
> > Thank you for the answers. I will raise my score to 6.

---

> > > ### Author Response · Authors · 2024-08-13
> > >
> > > Thank you for appreciating our work!

---

### Official Review · Reviewer_mQbj · 2024-07-12

**Soundness:** 3
**Presentation:** 2
**Contribution:** 3
**Rating:** 5
**Confidence:** 3

**Summary:**

This paper proposes a new algorithm, Manifold-LoRA, which incorporates the Stiefel manifold constraint to accelerate low-rank adaptation (LoRA) in fine-tuning LLMs. It also provides theoretical and experimental validation for the retraction-free and penalty parameter-free optimization methods.

**Strengths:**

This paper is highly technical and indicates a strong mathematical background in optimization and manifold theory. Manifold-LoRA leverages manifold geometry to reduce redundancy in LoRA fine-tuning, leading to enhanced performance and faster convergence. Furthermore, it has robust experimental validation across various datasets.

**Weaknesses:**

W1: Some experimental results are unclear and not well defined.

W2: Lack of the discussion about limitations of your method.

W3: Some findings of the experiments are hard to understand.

**Questions:**

Q1: In Table 1, could you please explain the meaning of m/mm? As I know, it's not an estimation of the performance of NLP models. You also didn't define Acc and Mcc. Also, why not use Acc to evaluate the performance of all the datasets?

Q2: The finding "It can be seen that our method 211 is consistently superior to other baselines." doesn't match the results in Table 1. What makes you conclude this?

Q3: Some findings are not clearly represented. For example, "We conclude that the proposed 228 Manifold-LoRA method achieves a 2x speed-up in training epochs compared to AdamW, while 229 simultaneously improving model performance".

Q4: The two datasets in the QA task are wildly used, but also weak. Have you tried other datasets? For example, the HotpotQA dataset.

**Limitations:**

Limitation is not enough and is meaningless.

---

> ### Author Rebuttal · Authors · 2024-08-06
>
> Thank you for reviewing our paper. The following are our responses.
> 1. **W1: Some experimental results are unclear and not well defined.**
>
>     **Reply**: We have revised the descriptions of numerical experiments accordingly. Specifically, we report the overall (matched and mismatched) accuracy for MNLI, Matthews correlation for CoLA, Pearson correlation for STS-B, and accuracy for other tasks. And for Table 2, we report Exact Match(EM) and F1 scores. Higher is better for all metrics.
>
> 2. **W2: Lack of the discussion about limitations of your method.**
>
>     **Reply**: In our conclusion, we address the theoretical limitations of our paper. While the constraints of the Stiefel and Oblique manifolds contribute to advancements in fine-tuning large language models, this method is difficult to extend to other areas such as pre-training and alignment. Therefore, it is essential to generalize our approach to more comprehensive manifolds, such as the Grassmann manifold. In addition, in the standard setting, LoRA uses a fixed rank for all layers, which has been shown to yield sub-optimal performance, as demonstrated in paper [46]. They address this issue by modifying the model architecture. In our conclusion, we also acknowledge that we did not use the coefficient matrix $A$ to dynamically select the rank $r$, which may have impacted our fine-tuning performance.
>
> 3. **W3: Some findings of the experiments are hard to understand.**
>
>     **Reply**: As shown in Table 1, our method outperforms the other baselines on average scores. Specifically, **Sphere (r=16)** achieves better performance than other baselines on 7 out of 9 tasks, with the exceptions of MNLI-MM and QQP. In Tables 2 and 3, our method performs consistently better than the baselines. We will make the statement about the experiments more precise.
>
> **Questions**:
> 1. **Q1**: In Table 1, could you please explain the meaning of m/mm? As I know, it's not an estimation of the performance of NLP models. You also didn't define Acc and Mcc. Also, why not use Acc to evaluate the performance of all the datasets?
>
>     **Reply**: MNLI-M and MNLI-MM represent two similar datasets for which the evaluation metric is accuracy. The MNLI dataset is a collection of sentence pairs annotated for textual entailment. MNLI-M is the matched split of the MNLI dataset, while MNLI-MM is the mismatched split.
>
>     Mcc stands for Matthews Correlation Coefficient, used for evaluating CoLA. The Pearson correlation is used for STS-B, and accuracy is the metric for other tasks. These metrics are widely used and are considered standard in the field, such as papers [22], [15], and [25].
>
> 2. **Q2**: The finding "It can be seen that our method 211 is consistently superior to other baselines." doesn't match the results in Table 1. What makes you conclude this?
>
>     **Reply**: The best results are in bold in Table 1, our method outperforms other methods in most datasets except in MNLI-MM and QQP compared to full fine-tuning. Also in the setting of Sphere(r=16), it achieves the best average scores among all methods, followed by Stiefel(r=8).
>
> 3. **Q3**: Some findings are not clearly represented. For example, "We conclude that the proposed 228 Manifold-LoRA method achieves a 2x speed-up in training epochs compared to AdamW, while 229 simultaneously improving model performance".
>
>     **Reply**: In Figure 2(a), it is easy to see that our method requires only half an epoch to reach a training loss of 1, whereas LoRA requires 2 epochs to achieve the same result. Additionally, since the additional computational cost per step induced by our method is negligible, the number of epochs needed can be roughly equated to the time required. In Figure 2(b) and Figure 2(c), it is obvious that our method achieves a higher EM (Exact Match) and F1 score.
>
> 4. **Q4**: The two datasets in the QA task are wildly used, but also weak. Have you tried other datasets? For example, the HotpotQA dataset.
>
>     **Reply**: Please refer to the following table for the results of HotpotQA dataset. Our method, Sphere $({r=16})$, performs the best.
>     | **Methods**       | **Params** | **HotpotQA (EM/F1)** |
>     |-------------------|------------|----------------------|
>     | Full FT           | 184M       | 63.3 / 76.7          |
>     | Adapter\({r=16}\) | 0.61M      | 60.2 / 74.2          |
>     | Bitfit            | 0.07M      | 57.2 / 71.6          |
>     | LoRA\({r=8}\)     | 1.33M      | 61.3 / 75.4          |
>     | LoRA\({r=16}\)    | 2.65M      | 61.5 / 75.4          |
>     | Sphere\({r=8}\)   | 1.33M      | 62.4 / 76.3          |
>     | Sphere\({r=16}\)  | 2.65M      | **63.4 / 76.9**      |
>     | Stiefel\({r=8}\)  | 1.33M      | 61.6 / 75.4          |
>     | Stiefel\({r=16}\) | 2.65M      | 61.4 / 75.4          |
>
>     **Table**: Results with DeBERTaV3-base on HotpotQA. We report EM/F1. The best results in each setting are shown in bold.

---

> > ### Comment · Reviewer_mQbj · 2024-08-13
> >
> > Thanks for your reply.
> >
> > 1. For Table 1, the design should clearly differentiate between dataset pairs and evaluators. For example, `m/mm` represents two similar datasets and should not be listed on the same line as evaluation metrics like Acc, Mcc, etc. Although Pearson Correlation is commonly used for the STS-B dataset, it is more appropriate to use Accuracy for the CoLA dataset, along with Mcc. Given that Accuracy is used for most datasets, it would be reasonable to present the performance of all methods for CoLA using Acc.
> >
> > 2. In your conclusion, instead of stating, "It can be seen that our method is consistently superior to other baselines," it would be more accurate to acknowledge that while your method outperforms other methods on most datasets, it does not do so on all datasets. Therefore, a more rigorous statement would be to highlight that your method demonstrates superior performance in the majority of cases.

---

> ### Author Response · Authors · 2024-08-13
>
> Thank you for your valuable feedback.
>
> For the Table 1, we will revise it to make it more clear.
>
> We use MCC for CoLA for two main reasons:
>
> 1. MCC is a widely recognized metric in the context of LoRA-fine-tuning. **It is considered a standard in LoRA fine-tuning due to the fact that the original LoRA paper[22](in their section 5.1 Table 2) reported MCC scores for the CoLA experiments in the GLUE benchmark**. This is the primary reason.
>
> 2. MCC is advantageous in handling imbalanced datasets because it evaluates all elements of the confusion matrix—true positives, true negatives, false positives, and false negatives—providing an unbiased assessment that accuracy (ACC) may not offer in skewed data situations. Moreover, MCC's scale from -1 to +1 provides detailed information about model performance, distinguishing between perfect predictions, random guesses, and complete inaccuracies, unlike ACC's more limited 0 to 1 range.
>
> We are, of course, willing to accommodate your suggestion by including the ACC scores for the CoLA dataset as well.
>
> We will also refine the statement "Our method is consistently superior to other baselines" to be more rigorous based on the results as you suggested.

---

> > ### Author Response · Authors · 2024-08-13
> >
> > The primary concerns you raised focus on the experimental section of our paper. We have clarified the metrics used in our experiments, which align with the standards of the filed. In addition, We have conducted experiments on the dataset as you suggested, and the results suggest the effectiveness of our method. Given that we addressed your primary concerns raised in the review, we would kindly ask you to adjust your review score while taking the discussion into account.

---

> > > ### Comment · Reviewer_mQbj · 2024-08-14
> > >
> > > Thank you for the answers. I will raise my score to 5.

---

### Author Rebuttal · Authors · 2024-08-07

**Continued rebuttal for Reviewer ohUo**

4. Without looking in detail in the proofs, my guess is that changing the penalty step size $\mu$ affects how close you need to be to the manifold (the value 1/8), which affects how small the step size need to be. In other words, the authors load all the complexity of setting the parameters onto the step size of the main objective. Saying there is an upper bound on its value, without specifying what that value is. You cannot call this parameter free.

    **Reply**: The step size bound $\alpha \in (0, \frac{1}{2c_1}]$ explicitly depends on the smoothness parameter and the bound of the gradient norm  of $f$, with $c_1$ defined in Line 418. It also implicitly depends on the penalty parameter and the neighborhood size determined by Lemma 4, which includes the condition $\alpha \le \frac{1}{24 \hat{D}_f}$ to ensure that iterates remain within the specified neighborhood. Generally, a smaller neighborhood size can lead to a faster convergence rate by Lemma 4 and allows a larger step size $\alpha$. However, $\alpha$ remains constrained by the landscape of the loss function, which may limit the allowable step size despite other conditions.

    **We should note that there are two parameters in landing algorithms: the step size and the penalty parameter.** Specifically, the landing fields in both [1] and our manuscript take the form: $ V(X) = \alpha \nabla f(X) + \mu \varphi(X) $. In the previous paper [1], both parameters were unknown, requiring a sufficiently large $\mu$ and a varying $\alpha$ (with respect to the iteration number) to achieve convergence. Instead, by leveraging the strong convexity-like property of the penalty problem, we demonstrate convergence using $\mu = \frac{1}{3}$ and a constant step size $\alpha$. In this sense, we describe our method as penalty parameter-free.

5. Another point is that the values of the parameter probably affect the convergence rate, though the authors do not discuss this at all.

    **Reply**: The proof of Theorem 1 establishes that $\sum_{k=0}^K \alpha \lVert grad f(X_k)\rVert^2 < \infty$. This implies $\min_{k =0, \ldots, K} \lVert grad f(X_k)\rVert^2 \leq \mathcal{O}(\frac{1}{\alpha K})$. With this bound, a larger step size leads to faster convergence. This complexity bound aligns with the best-known results for retraction-based methods. We will revise our manuscript to state this clearly.

### Minor comments (do not affect the recommendation):

- Line 117: If X is on the manifold, shouldn't $\bar{X}$, which is the projection of X on the manifold, be exactly X?

    **Reply**: $X$ should be any matrix of $\mathbb{R}^{n\times p}$ and $\bar{X}$ is its projection on ${\rm St}(n,p)$. We have revised it accordingly.

- Line 119: "satisfies the restricted secant"

    **Reply**: Revised.

- Line 131: What is $U_{\rm St} (1/8)$ ? Not defined. Ditto line 136.

    **Reply**: In line 131, $U\_{\rm St}(1/8) = \\{ Y \in \mathbb{R}^{n \times p} \mid \lVert Y - X \rVert_F < \frac{1}{8} \\}$.
    In line 136, it should be $U_{\rm St}(1/8)$. Sorry for the confusion. We have added this in the notation part.

- Line 133: If the condition of twice diff is assumed, then state it earlier.

    **Reply**: Revised.

- Eq (10): $\hat{D}\_f$ is not defined.

    **Reply**: In Line 483, we define $\hat{D}\_f$ as $
    \hat{D}\_f := \max_{X \in \bar{U}_{\text{St}(d,r)}\left(\frac{1}{8}\right)} \lVert\nabla f(X)\rVert
    $

- Table 1: Why metrics are changing between columns?

    **Questions**:

    **Reply**: MNLI-M and MNLI-MM represent two similar datasets for which the evaluation metric is accuracy. The MNLI dataset is a collection of sentence pairs annotated for textual entailment. MNLI-M is the matched split of the MNLI dataset, while MNLI-MM is the mismatched split.

    Mcc stands for Matthews Correlation Coefficient, used for evaluating CoLA. The Pearson correlation is used for STS-B, and accuracy is the metric for other tasks. These metrics are widely used and are considered standard in the field, such as papers [22], [15], and [25].

### Questions:

1. Please address more carefully why you think you are novel with respect to [1]?

    **Reply**: See response to W1

2. Why are fast retraction methods not sufficient?

    **Reply**: See response to W1

3. How setting the parameters affect convergence rate?

    **Reply**: See response to W3

---

### Decision · Program_Chairs · 2024-09-25

**Decision:**

Reject

**Comment:**

This meta review is based on the reviews, the authors responses and post-rebuttal discussions, and ultimately my own judgement on the paper. The reviewers are mostly positive about this submission and unanimously recommend it for borderline acceptance or weak acceptance. It was agreed in the reviews that the proposed retraction-free first-order algorithm is simple yet effective for solving a regularized variant of the Stiefel manifold optimization problem, and its application to LoRA for fine-tuning large language models sounds very interesting. In their initial review, the reviewers also expressed a number of concerns regarding the discrepancy between theory and application, setting of algorithm hyper-parameters, and explanation of empirical results as well, which appear to be resolved by the authors responses. However, during the reviewers discussion phase, a severe technical flaw in the proof of Theorem was identified as specified below:

In the last but one inequality in (17), we believe the authors have used the self-adjointness $\langle grad f(X_k), \nabla \phi(X_k)\rangle = \langle \nabla f(X_k), P_{T_{X_k }St(d,r)}(\nabla \phi(X_k))\rangle$, of which the validness seems highly non-trivial, if not impossible, to show for the iterates $X_k\notin St(d,r)$ produced by the proposed retraction-free algorithm. Unfortunately, this passage was completely left unjustified in the block of comments below the proof.

Since Theorem 1 lays the foundation on the landing convergence of algorithm, the above mathematical flaw could be fatal to substantially ruin the contribution of this work. In addtion, the entire convergence analysis part, Section 2.3, reads somewhat rough (see Lemma 4 and its proof arguments for another instance).

Overall, I found that the weaknesses outweight the strengths of this paper. A major improvment for correcting the proof of Theorem 1 is mandatory should the paper be considered for publication, which however is estimated to be out of the scope of a camera-ready update. In view of this, it was eventually decided not to accept this paper in the current form. We do believe this paper, once carefully revised, will get published somewhere, and hope the authors would find the reviews helpful for future submission.